

# The IPCC Sixth Assessment Report WGIII climate assessment of mitigation pathways: from emissions to global temperatures

Jarmo S. Kikstra[1,2,3], Zebedee R.J. Nicholls[1,4,5], Christopher J. Smith[1,6], Jared Lewis[1,4,5], Robin D. Lamboll[2,3], Edward Byers[1], Marit Sandstad[7], Malte Meinshausen[4,5], Matthew J. Gidden[1,8], Joeri Rogelj[1,2,3], Elmar Kriegler[9,10], Glen P. Peters[7], Jan S. Fuglestvedt[7], Ragnhild B. Skeie[7], Bjørn H. Samset[7], Laura Wienpahl[1], Detlef P. van Vuuren[11,12], Kaj-Ivar van der Wijst[12], Alaa Al Khourdajie[3], Piers M. Forster[6], Andy Reisinger[13], Roberto Schaeffer[14], Keywan Riahi[1,15]

[1]Energy, Climate and Environment (ECE) Program, International Institute for Applied Systems Analysis (IIASA), Laxenburg, 2361, Austria
[2]The Grantham Institute for Climate Change and the Environment, Imperial College London, London, UK
[3]Centre for Environmental Policy, Imperial College London, London, UK
[4]Climate & Energy College, School of Geography, Earth and Atmospheric Sciences, The University of Melbourne
[5]Climate Resource, Melbourne, Australia
[6]Priestley International Centre for Climate, University of Leeds, Leeds, United Kingdom
[7]CICERO Center for International Climate Research, Oslo, Norway
[8]Climate Analytics, Berlin, Germany
[9]Potsdam Institute for Climate Impact Research (PIK), Potsdam, Germany
[10]Faculty of Economics and Social Sciences, University of Potsdam, Potsdam, Germany
[11]PBL Netherlands Environmental Assessment Agency, The Hague, The Netherlands
[12]Copernicus Institute of Sustainable Development, Utrecht University, Utrecht, The Netherlands
[13]Institute for Climate, Energy and Disaster Solutions, Fenner School of Society & Environment, Australian National University, Canberra, Australia
[14]Centre for Energy and Environmental Economics (CENERGIA), COPPE, Universidade Federal do Rio de Janeiro (UFRJ), Rio de Janeiro, Brazil
[15]Graz University of Technology, Graz, Austria

*Correspondence to*: Jarmo S. Kikstra (kikstra@iiasa.ac.at)





**Abstract.**

While the IPCC's physical science report usually assesses a handful of future scenarios, the IPCC Sixth Assessment Working Group III report (AR6 WGIII) on climate mitigation assesses hundreds to thousands of future emissions scenarios. A key task is to assess the global-mean temperature outcomes of these scenarios in a consistent manner, given the challenge that the emission scenarios from different integrated assessment models come with different sectoral and gas-to-gas coverage and cannot all be assessed consistently by complex Earth System Models. In this work, we describe the "climate assessment" workflow and its methods, including infilling of missing emissions and emissions harmonisation as applied to 1,202 mitigation scenarios in AR6 WGIII. We evaluate the global-mean temperature projections and effective radiative forcing characteristics (ERF) of climate emulators FaIRv1.6.2, MAGICCv7.5.3, and CICERO-SCM, discuss overshoot severity of the mitigation pathways using overshoot degree years, and look at an interpretation of compatibility with the Paris Agreement. We find that the lowest class of emission scenarios that limit global warming to "1.5°C (with a probability of greater than 50%) with no or limited overshoot" includes 90 scenarios for MAGICCv7.5.3, and 196 for FaIRv1.6.2. For the MAGICCv7.5.3 results, "limited overshoot" typically implies exceedance of median temperature projections of up to about 0.1°C for up to a few decades, before returning to below 1.5°C by or before the year 2100. For more than half of the scenarios of this category that comply with three criteria for being "Paris-compatible", including net-zero or net-negative greenhouse gas (GHG) emissions, are projected to see median temperatures decline by about 0.3-0.4°C after peaking at 1.5-1.6°C in 2035-2055. We compare the methods applied in AR6 with the methods used for SR1.5 and discuss the implications. This article also introduces a 'climate-assessment' Python package which allows for fully reproducing the IPCC AR6 WGIII temperature assessment. This work can be the start of a community tool for assessing the temperature outcomes related to emissions pathways, and potential further work extending the workflow from emissions to global climate by downscaling climate characteristics to a regional level and calculating impacts.

**Short summary (500 character).**

Assessing hundreds or thousands of emission scenarios in terms of their global-mean temperature implications requires standardised procedures of infilling, harmonisation and probabilistic temperature assessments. We here present the 'climate-assessment' workflow that provides the methodology used in the IPCC Working Group III report.

**1 Introduction**

The Working Group III (WGIII) contribution to the Intergovernmental Panel on Climate Change (IPCC) Sixth Assessment Report (AR6) assesses the recent literature on how climate change can be mitigated (IPCC, 2022c). A key part of this assessment uses emissions scenarios (Riahi et al., 2022) that explore a variety of climate change mitigation futures. The Paris Agreement, which specified a long-term global temperature goal (UNFCCC, 2015), strengthened by the Glasgow Climate Pact



stressing the 1.5°C temperature level (UNFCCC, 2021), made it ever more relevant to determine global mean surface
temperature outcomes in assessments of policy-relevant climate mitigation literature.

In this paper, we (a) lay out and discuss the methodology used in IPCC AR6 for assessing the global warming

implications of scenarios with sufficient information about their emissions, (b) describe the global mean temperature outcomes
of the scenario set available in the AR6 WGIII report's scenarios database (AR6DB, Byers et al., 2022), and (c) document and
provide the tools used for this part of the assessment. These temperature projections from IAM scenarios are used across many
parts of the WGIII report. This paper provides a further detailed description of the climate assessment workflow that was used
in a few sections in the Summary for Policymakers (SPM) (IPCC, 2022d), and especially in Chapter 3 on Mitigation Pathways
Compatible with Long-Term Goals (Riahi et al., 2022), with a summary of the methods and some additional analysis in Annex
III on Scenarios and Modelling Methods (IPCC, 2022a).

A comprehensive assessment of the global temperature outcomes of long-term greenhouse gas (GHG) emissions scenarios
requires diverse emissions data to be made comparable, gaps in data to be completed, and tools to project global temperature
from those emissions that reflect the best available climate science knowledge. After a selection of scenarios that comply with
reporting standards and are within ranges of uncertainty ("vetting") is made, global mean temperature outcomes are calculated.
The climate assessment workflow we describe here has three core steps: 1) *harmonisation* of emissions, 2) *infilling* of
emissions, 3) running one or several emissions-driven reduced-complexity climate models (see **Figure 1**).

In the *harmonisation* process, scenarios are made comparable by ensuring they start from the same historical emission

levels. This ensures that differences between climate futures resulting from two different pathways are the result of future
emissions due to structural change in mitigation scenarios rather than different historical emissions estimates or assumptions.

In the *infilling* step, data gaps in emissions scenarios, such as time evolutions for some individual gas or aerosol

species that are not reported by a given integrated assessment model (IAM), are closed by inferring representative trajectories
of those missing species from the wider literature.

In the *climate* run step, reduced complexity climate models (also known as climate emulators) are used to project the

physical climate response to emissions. These climate emulators are calibrated to closely reproduce historically observed
warming, projections of warming for standard scenarios, and the uncertainty ranges in key physical climate parameters
assessed in the IPCC Working Group I report (IPCC, 2021). This close collaboration between WGI and WGIII to ensure
consistency of climate assessments across various IPCC AR6 products is a key development compared to the IPCC Fifth
Assessment Report (AR5) (IPCC, 2014, 2013) and earlier IPCC Assessment Reports. The AR6 WGIII report is the first IPCC
report that uses climate emulators that are fully in line with complex models and other lines of evidence as assessed by the
physical science basis of the same cycle.

A total of 3,131 global and regional scenarios were submitted to the AR6 Scenario Explorer hosted by IIASA (Byers

et al., 2022). Out of this set, 1,686 global scenarios were considered to meet minimum quality standards for use in long term





scenarios assessment based on the vetting criteria as set out in Annex III of IPCC WGIII. This set was further narrowed down
to 1,202 scenarios (IPCC, 2022a) that contained sufficient emission data across gases and sectors to provide full-century
climate outcomes. This sub-selection to more complete scenarios ensures that the harmonised and infilled emissions reflect
the intention of the prospective modelling in the original scenario submission. For the main text, figures, and tables in this
paper, we use this set of 1,202 scenarios.
In the remainder of this paper, we start by placing the IPCC WGIII AR6 infilling steps, harmonisation procedures
and climate assessment in its historical context and present the criteria it aimed to meet. Then, we provide details on the
methods applied going from emissions provided by IAMs to output from climate emulators. Lastly, we touch upon future
development options.
**2 History of scenario temperature projections in IPCC WGIII reports and the updated process in AR6**
**2.1 History of climate assessment processes**
**2.1.1 Climate emulators in IPCC reports**
Climate emulators have been used by the IPCC from its very start. For instance, the First Assessment Report explains that
"simpler models, which simulate the behaviour of [General Circulation Models (GCMs)], are also used to make predictions of
the evolution with time of global temperature from a number of emission scenarios. These so-called box-diffusion models
contain highly simplified physics but give similar results to GCMs when globally averaged." (IPCC, 1992). Emulators, because
of their computational simplicity, can be used much more widely than complex GCMs or Earth System Models (ESM).
Because of limited ability in 1990s to perform long-term coupled atmosphere-ocean runs with a broad coverage of different
greenhouse gases and aerosols and an interactive carbon cycle, the early assessment reports relied heavily on simple climate
models, including the Working Group I reports. A technical overview report about their strength and limitations was published
by the IPCC in 1997 (Houghton et al., 1997). With an increasing availability of Earth system models of intermediate
complexity (EMICs), coupled atmosphere-ocean general circulation models (AOGCMs) and ultimately the fully-fledged Earth
system models (ESMs), the focus shifted in the physical WGI reports towards the use of progressively more complex models.
However, in the AR6 Working Group I (WGI) report, climate emulators were used to fill in gaps from experiments of interest
that are not run by ESMs (e.g. SPM figs. 2c and 4b), and also to bridge the gap between expert assessment of the climate
system and some of the unconstrained projections resulting from ESMs (Hausfather et al., 2022; Lee et al., 2021; Forster et
al., 2021). Multiple lines of evidence in support of the assessment of climate sensitivity and other climate characteristics led
to IPCC WGI AR6 adopting a new approach, which also involved calibrating climate emulators to translate the assessment of
key climate characteristics into the global-mean temperature projections. Additionally, the increased focus on translating





insights from WGI to other stakeholders and scientific communities included stronger cross-WG collaboration and triggered
a concerted effort for climate emulator calibration on the basis of a wide range of WGI assessment results.

In the Working Group III report, there are two key reasons for using climate emulators to assess the temperature outcomes

of long-term climate mitigation scenarios. The first reason is time and resources. With a large number of scenarios available
from a wide variety of studies, it would take too much computing time to rapidly simulate all scenarios by one ESM, let alone
by a wider set of models such as those that participate in international initiatives like the Coupled Model Intercomparison
Project (CMIP). For instance, a quick turnaround was required between WGIII's literature cut-off date (11 October 2021) by
which scenarios had to be confirmed as published, and its deadline for Final Government Draft submission by authors (1
November 2021). It is computationally not feasible for modern ESMs to run all scenarios in this timespan. Typically, an IPCC
report undergoes multiple expert and government reviews. This means that the climate assessment is repeated multiple times
over the course of an IPCC report drafting cycle, which for AR6 WGIII AR6 took 3 years from the first lead author meeting
to the approval of the SPM. The second reason is mirroring the reasoning in Working Group I for the use of climate emulators
to combine multiple lines of evidence to represent the overall best estimate and uncertainty range. In the Working Group III
context, a single ESM, or even a set of them, is unlikely to match the best estimate as well as physical climate uncertainty of
the assessed temperature response to anthropogenic emissions with a good representation of uncertainty as assessed by WGI,
or even necessarily reproduce historically observed global mean temperatures well (Smith and Forster, 2021).

**2.1.2 Long-term mitigation pathway assessments in previous IPCC WGIII reports**

This exercise sits within a tradition of large-scale assessments and previous IPCC WGIII reports, though the practice to group
mitigation scenarios based on climate emulator outcomes is more recent. Using two models, the First Assessment Report
(FAR) WGIII (Houghton et al., 1990) evaluated 3 mitigation scenarios (SA90) and two reference scenarios and their
atmospheric $CO_2$ and $CO_2$-equivalent concentrations but did not directly assess global temperature outcomes related to these
scenarios. The 1992 supplement to the FAR (IPCC, 1992) evaluated six alternative emissions scenarios (IS92 a-f) and provided
global warming estimates, using the best estimate of climate sensitivity available at that time. In a 1994 follow-up report, the
radiative forcing characteristics of the IS92 pathways were assessed in much more detail (IPCC, 1994). The Second
Assessment Report (SAR) (IPCC, 1996) assessed a wider range of socioeconomic scenarios and used a more extensive set of
simple climate models (Houghton et al., 1997), but did not use these to assess the temperature implications of the mitigation
scenario literature. In similar fashion, WGIII of the Third Assessment Report (TAR) (IPCC, 2001) also did not perform its
own temperature assessment or grouping of mitigation scenarios by climate categories but used $CO_2$ concentrations as
stabilisation levels for the assessment of the mitigation pathways (e.g. SPM.1 and Table 2.6 in IPCC WGIII TAR).

The WGIII Fourth Assessment Report (AR4) contained the first IPCC temperature assessment of emissions scenarios

from the available literature. 177 scenarios were assessed, covering a mix of $CO_2$-only and multi-gas studies. Scenario
characteristics were compared by grouping them in six categories, based on climate targets as reported in each of the original
peer-reviewed articles assessed by the IPCC. Where data was unavailable, scenario characteristics for either $CO_2$




concentrations or radiative forcing within each category (15th and 85th percentile) were derived using the relationship between
$CO_2$ concentrations and radiative forcing, and the relationship between $CO_2$ concentrations and equilibrium temperature. Only
6 scenarios fell in the lowest warming category, which was associated with 2.5-3.0 W/m² radiative forcing and $CO_2$
concentrations of 350-400 ppm in 2100, with a rough estimate of 2.0-2.4°C global mean surface temperature increase above
pre-industrial levels (here referring to the era before the industrial revolution of the late 18th and 19th centuries, while in the
rest of the paper we refer to the period from 1850-1900) *at equilibrium* using a climate sensitivity of 3°C per doubling of $CO_2$
concentrations. The highest category covered the 6.0-7.5 W/m² range of forcing and featured only 5 scenarios. The report was
clear about the limitation of this approach, writing in subsection 3.3.5 that "it should be noted that the classification is subject
to uncertainty and should thus be used with care" (IPCC, 2007).
In the Fifth Assessment Report (AR5) WGIII report, a larger database of 915 scenarios was available for the
assessment of mitigation pathways. These scenarios differed in their design (e.g., ever-growing emissions, climate stabilisation,
or peak-and-decline scenarios), as well as in how many gases were included. Despite the methodological difficulties in
comparing multiple types of scenarios, AR5 still grouped scenarios in different climate categories to enable comparison of
their key characteristics (IPCC, 2014). With the scenario literature at that time often using 2100 radiative forcing targets to
design scenarios, including the Representative Concentration Pathways (RCPs), CO2-equivalent concentrations in 2100 were
chosen as a classification indicator (CO$_2$-equivalent concentrations represent the concentration of carbon dioxide ($CO_2$) that
would cause the same radiative forcing as a given mixture of $CO_2$ and other forcing components). The calculation of $CO_2$-
equivalent concentrations in 2100 from emissions was standardised. All scenarios with at least information on total Kyoto gas
emissions were assessed using the climate emulator Model for the Assessment of Greenhouse Gas Induced Climate Change
(MAGICC) version 6.3 (Meinshausen et al., 2011b, a). This model version drew on a probabilistic ensemble of which
concentration and radiative forcing outcomes were constrained by observations and physical climate parameter uncertainties
assessed in AR4 (Meinshausen et al., 2009; Schaeffer et al., 2013), with model updates to better reflect the climate sensitivity
distribution as assessed in AR5 WGI (Rogelj et al., 2012). To group scenarios, the median CO$_2$-equivalent concentration of
total radiative forcing of this probabilistic ensemble was used. For emissions harmonisation, to avoid artefacts in the
temperature projections resulting from differences in model-reported and historical emissions, emissions were set to historical
observation values in 2010, with the difference to model-reported values linearly declining to zero in 2050 (Krey et al., 2014).
At minimum $CO_2$ from the energy and industrial processes (E&IP) sector (also known as $CO_2$ from the use of fossil fuels and
industry (or CO$_2$-FFI, as used in AR6)), and $CH_4$ and $N_2O$ from E&IP and land use sectors from each individual scenario
needed to be available. For emissions infilling of other species, a set of heuristics was applied to fill in any missing F-gas,
carbonaceous aerosols, and/or nitrate emissions (Krey et al., 2014). Another set of practical heuristics was developed to classify
scenarios that did not report all necessary greenhouse gas and other emissions or did not report emissions until the end of the
21st century. The classification of such scenarios into groups was based on only Kyoto gas forcing (given a lack of total forcing)
in 2100, cumulative $CO_2$ emissions from 2011 to 2100, and cumulative $CO_2$ emissions from 2011 to 2050, in order of



preference. One hundred and fourteen scenarios were classified in the lowest category of 2.3-2.9 W/m$^2$ in 2100, with associated
2100 median temperatures ranging from 1.5 to 1.7°C above 1850-1900 levels.

The Special Report on Global Warming of 1.5°C (IPCC, 2018) – abbreviated as SR1.5 – featured an extensive climate

assessment of emissions scenarios with the most advanced methods so far. After the introduction of temperature targets in
international climate policy in the Cancún Agreement of 2010 (UNFCCC, 2010), and the subsequent adoption of the Paris
Agreement with its specific long-term temperature goal a stated in Article 2 of the agreement (UNFCCC, 2015), SR1.5 was
the first IPCC report where scenarios were categorised based directly on their projected global temperature outcomes. This
temperature categorisation followed the practice established by the Emissions Gap Reports series of the UN Environment
Programme (Hare et al., 2010; Rogelj and Shukla, 2012; Rogelj et al., 2011). SR1.5 only assessed scenarios with information
until 2100 for at minimum $CO_2$ from E&IP and (total) $CH_4$, $N_2O$, and sulphur dioxide emissions. The SR1.5 approach used
the same harmonisation method as AR5, but because an absolute offset harmonisation method would have turned some non-
$CO_2$ emissions pathways negative, SR1.5 rather used a multiplicative ("ratio") method (Forster et al., 2018). For the infilling
of emissions species not reported, including F-gases and black carbon (BC), values from the low forcing scenario RCP2.6 (van
Vuuren et al., 2011; Meinshausen et al., 2011a) were used, in line with the focus of the report on 1.5°C and 2°C consistent
scenarios. A total of 368 scenarios (out of 529 submitted scenarios) were grouped into six temperature categories, five of which
were to indicate different categories of below 2°C scenarios (Forster et al., 2018; Rogelj et al., 2018: Table 2.4 and Table
2.SM.12). Using a MAGICC6 setup similar to that used in AR5 (Meinshausen et al., 2011a, b; IPCC, 2014), temperature
exceedance probability at peak temperature and in 2100 were used to define these categories. In addition, the climate emulator
Finite Amplitude Impulse Response (FaIR) version 1.3 (Smith et al., 2018) was used to run all scenarios for a sensitivity
analysis. FaIRv1.3 and MAGICC6 produced substantially different temperature and forcing levels for the same emissions
scenarios, with FaIRv1.3 typically projecting less warming, and MAGICC6 more, mostly due to effective radiative forcing
from non-CO2 components. MAGICC6 was used for the main classification because it was more established in the literature,
provided direct comparability of AR5 in absence of a more recent IPCC WGI assessment, and had been tested against CMIP5
models (Forster et al., 2018, 2SM-3).

AR6 for the first time in IPCC WGIII assessments used a fully integrated temperature-based classification of

mitigation scenarios, with the climate emulators used in WGIII being fully consistent with WGI of the same assessment cycle
following extensive calibration and testing exercise of emulators to assess their suitability to reproduce assessed climate ranges
(Forster et al., 2021). The use of climate emulators in WGIII was motivated by several considerations.

Firstly, the main physical reason for using a radiative forcing-based measure over temperature in earlier reports,

namely an uncertain climate sensitivity (Krey et al., 2014, page 1312 of AR5 WGIII), has been ameliorated by much more
robust constraints on both equilibrium climate sensitivity (Sherwood et al., 2020) and the transient climate response (Forster
et al., 2021). This allows a more robust estimate of the temperature response from a given emission pathway.





Secondly, there was considerable ambiguity in earlier assessments about which forcing agents were included in the
radiative forcing classification as sometimes total anthropogenic forcing estimates (or subsets thereof) were used and
sometimes only GHGs were included.
Thirdly, the "$CO_2$ equivalent concentration" classification in earlier reports created some confusion for readers in the
context of the more widely used, but rather different concept of "$CO_2$ equivalent emissions".
Finally, and most importantly, the Paris Agreement long-term global temperature goal means that a global
temperature classification of emission scenarios is now directly relevant to inform policy decisions.

## 2.2 Design criteria for a new process

The development of this workflow builds on experience from previous IPCC reports. In broad lines, IPCC AR6 WGIII
followed the methodology as applied in SR1.5, while addressing multiple outstanding issues and knowledge gaps. These
include (a) increased reproducibility, openness, and transparency, (b) usage of multiple consistently calibrated and extensively
evaluated climate emulators, and (c) more advanced methods to represent non-$CO_2$ emissions and forcing.

### 2.2.1 Reproducibility, openness, and transparency

During the preparation of AR6, accessibility and reproducibility of scientific results were identified as a key aspect to be
addressed in the production of the report. This relies on the transparency and reusability of the products and tools underpinning
the production of these scientific results (Iturbide et al., 2022).
The long-term global emissions pathways literature largely relies on IAMs, an increasing number of which are
becoming accessible via open-source codes and training material for potential users (Skea et al., 2021). In the WGIII report,
an increased attention has gone into documenting the core assumptions and characteristics of IAMs in order to facilitate the
interpretation and reproducibility. These pathways have been published in peer-reviewed articles, and none of them are created
by the IPCC itself. What is however done for the IPCC assessment report is the consistent comparative analysis of the
temperature outcomes of the different scenarios based on their emissions.
Until now, the climate assessment process utilised by the IPCC has been described in the report, but never discussed
in detail or been made openly available to the community as a software tool. Making the climate assessment process open-
source will not only facilitate the reproducibility of the report's scientific findings, but also facilitate future analyses of new
data applying a methodology consistent with the AR6 WGIII report.
Doing so is in line with previous efforts such as in AR5 and SR1.5, where the scenario data and climate assessment
information were made accessible in a format following community standards (Huppmann et al., 2018b, a; IIASA, 2014). In
addition, increased transparency was provided by releasing the calculations to get from the scenario data to the presented
figures and tables in SR1.5 (Huppmann et al., 2018c). A growing body of research has developed describing analyses that
compare emissions pathways and their temperature outcomes including climate-policy target quantification (Höhne et al.,
2021; Meinshausen et al., 2022) and grey-literature mitigation scenario assessment (Brecha et al., in preparation).





### 2.2.2 The inclusion of multiple climate emulators

The two emulators used in SR1.5 exhibited substantial differences in the near-term warming and it was unclear how much of these differences were structural and how much was from different calibrations (Forster et al., 2018). Since then, emulator diversity and the understanding of differences between emulators have improved. Structural uncertainties have been probed by comparing idealised simulations of a range of emulators with different physical characteristics all run with the same best-estimate climate sensitivity (Nicholls et al., 2020). Emulators were able to simulate global mean surface temperatures of more complex models within a root-mean-square error of 0.2 ℃ over a range of experiments across a range of scenarios. As the ESMs themselves have structural differences, the emulator with the best fit to a given ESM varied. Because it is not known which ESM best captures reality, these results present an inherent structural uncertainty. This structural uncertainty is therefore best explored by using a diverse range of emulators to assess the climate response across scenarios. Diversity comes from both how emulators capture the emissions to radiative forcing relationship across considered emissions and from how the transient surface temperature response to a given forcing is represented. To allow for a multi-model assessment, four emulators were calibrated to the same set of WGI AR6 physical responses (Forster et al., 2021). The calibration approach varied amongst the emulators (Smith et al., 2021). Nevertheless, they produced a similar best estimate and range of responses to the assessment they were trying to match. Newly developed techniques (Nicholls et al., 2021a) were applied to evaluate the probabilistic distributions of each emulator. Based on these techniques, WGI concluded that FaIRv1.6.2 and MAGICCv7.5.3 were generally able to match the best estimates of multiple climate indicators, including the change in global mean surface temperature to within 5% and match the *very likely* ranges to within 10% (Forster et al., 2021).

### 2.2.3 Increased detail for non-CO₂ greenhouse gases and aerosols

$CO_2$ is the dominant driver of long-term global climate change, but non-$CO_2$ GHG emissions and aerosols play a significant role on different time scales and reducing warming from non-$CO_2$ related emissions is important to meet climate targets. IPCC WGI (IPCC, 2021b) found that historical $CO_2$-induced warming was 0.8℃ (1850-1900 to 2010-2019), while methane-induced warming was 0.5℃ and sulphate aerosol-induced cooling 0.5℃, with additional changes from other emission components and sources. Therefore, while cumulative $CO_2$ is the strongest determinant of temperature outcomes, non-$CO_2$ emissions pathways including short-lived climate forcers (SLCFs) are important in analysing how different temperature projections of scenarios compare (Damon Matthews et al., 2021; Samset et al., 2020; Rogelj et al., 2015, 2018; Allen et al., 2009).

Historically, IAMs have predominantly focussed on modelling $CO_2$ emissions, with other major GHG emissions like methane receiving less attention. Other emissions including minor GHGs, aerosols, and aerosol precursors are covered by fewer models. Some gases that are represented in climate emulators are not modelled for any long-term global scenario IAM considered in AR6, though these particular emissions have very small projected impact on climate change. To maximise the richness and diversity of scenarios available in a given assessment (Guivarch et al., 2022), a process of infilling scenarios with missing emission data is performed. There is, however, no unique way to infill scenarios with missing data.





Previous assessments (section 2.1) already undertook a process of infilling, but due to limited available peer-reviewed
literature and tools, these methods were rather simple and did not include emissions species methods or scenario-specific
infilled pathways. As an example, in IPCC SR1.5, missing data was taken from SSP1-2.6, on the basis that the assessment was
focused on 1.5°C and 2°C scenarios rather than the full range including baseline scenarios. This means that there can be an
inconsistency between infilled and original IAM emissions in terms of the implicit underlying socio-economic drivers or
compound emissions.  Particularly in the short term, SLCFs can have a significant effect on temperature. With new literature
and tools available (Lamboll et al., 2020), the AR6 WGIII scenario workflow adopted a more systematic approach to infilling
that captures more detail in non-CO$_2$ emissions of scenarios (IPCC, 2022a).

**3 Methods**

The climate assessment workflow as visualised in Figure 1 was implemented using the Python programming language (Van
Rossum and Drake Jr, 1995), and is available as an open-source Python package from https://github.com/iiasa/climate-
assessment (Kikstra et al., 2022a), with detailed documentation available at: https://climate-assessment.readthedocs.io.

**3.1 Scenario vetting**

Global scenarios used to assess climate mitigation options in were extensively vetted to ensure minimum reporting of relevant
variables and check that reported values in the model base years fall within ranges of uncertainty as specified in
**Supplementary Table 1**. Whilst IAMs report a large number of sectoral variables, for the purposes of this assessment the
vetting was limited to global emissions and energy related variables. This process was repeated during the call for scenarios
such that model teams had the opportunity to review the results of the vetting process, diagnose results and correct reporting
errors. As a minimum, IAM teams needed to report global emissions for CO$_2$, CH$_4$, N$_2$O through the period 2015 to 2100 for
inclusion of a scenario in the temperature assessment. Checks on variables for specific technologies were also made for nuclear,
CCS, solar and wind power as well as primary energy. For emissions, interpolated modelled emissions for 2019 were checked
against the 2019 values from two emissions data sets (Minx et al., 2021; Nicholls et al., 2021a). For the detailed values used
for the thresholds applied, see **Supplementary Table 1**.
From 2266 global scenarios considered in the AR6 Scenarios Database with at least a relevant emissions or energy
variable, about three quarters passed the energy and emissions criteria, whilst only 1202 passed all vetting criteria and
minimum emissions reporting requirements. The most exclusionary criteria were those for nuclear and solar and wind
electricity production in 2020, where for each criterion 266 and 377 scenarios were out of range, respectively.

**3.2 Harmonisation of emissions pathways**

Emissions harmonisation refers to the process used to align modelled GHG and air pollutant pathways with a common source
of historical emissions. This capability enables a common climate estimate across different models, increases transparency and




robustness of results, and allows for easier participation in intercomparison exercises by using the same, openly available
harmonisation mechanism (Gidden et al., 2019). In the AR6 climate assessment workflow the open-source Python software
package called 'aneris' (Gidden et al., 2018) was used for harmonisation.

In principle, many methods to align modelled results with historical emissions could be used. In past IPCC

assessments, ratio (multiplicative) methods (AR5) and offset (additive) methods (SR1.5) have been employed. Gidden et al.,
(2018) introduced a common approach for choosing which methods should be applied in different contexts (the so-called
'default decision tree'). In AR6, this approach was used where suitable. For some species, however, a specific method was
chosen in AR6 (see Table 1 for the full overview). For $CO_2$-FFI, a ratio-based method was used with convergence in 2080, in
line with the application of aneris for the CMIP6 process (Gidden et al., 2019). The convergence for 2080 is later than in
SR1.5, which used 2050 (Forster et al., 2018). A later convergence year was seen as more suitable when considering scenarios
across a wider range of mitigation futures than was considered in SR1.5. For $CO_2$ from AFOLU, an offset method with a
convergence target in 2150 was used as the preferred method to deal with high historical interannual variability and large
uncertainty in historical emissions estimates (Dhakal et al., 2022) leading to similarly large differences in historical emissions
estimates from separate IAMs (IPCC, 2022d). All other emissions species with high historical variance are harmonised using
a ratio method with a convergence target in 2150. Remaining F-gases are harmonised at the individual species level, increasing
the detail compared to SR1.5, but because of low model reporting confidence a constant ratio harmonisation method is used.
For all other emissions species, we use the default settings of (Gidden et al., 2018, 2019).

*For harmonisation,* AR6 WGIII used the same historical emissions that were also used for the emissions-driven

CMIP6 (Gidden et al., 2019) and RCMIP (Nicholls et al., 2020, 2021a) emissions-driven runs. This dataset is a combination
of historical emissions databases. A significant share comes from the Community Emissions Data System (CEDS) database
(Hoesly et al., 2018), but additional sources and methods have been used (for full detail, see Nicholls et al., (2020, 2021a),
Gidden et al., (2019), and Kikstra et al., 2022a). The year of 2015 was taken for harmonisation in line with CMIP6. In the case
that IAM-reported values are not available for 2015, but did report modelled 2010 and 2020 emissions, the difference from
historical data in 2010 was used to infer a 2015 value before harmonising. The benefit of using a similar dataset and methods
as for emissions-driven CMIP6 and RCMIP which informed the assessment by WG I is that this leads to high consistency of
modelled temperature outcomes for emissions scenarios assessed by WGIII with the assessment of physical climate science
by WGI, and thus a stronger coherence across IPCC Working Groups within the AR6.
**3.3 Infilling of emissions pathways not reported by scenarios submitted to AR6 database**
If for instance a modelled scenario reports most climate relevant species but not black and organic carbon, which are required
by climate emulators to project temperature outcomes, the infilling process will supplement the model reported results with
synthetic estimates of black and organic carbon. Infilling thus ensures that all climate-relevant anthropogenic emissions are
included in each climate run for each scenario. This makes the climate assessment of alternative scenarios more comparable
and reduces the risk of a biased climate assessment, because not all climatically active emission species are reported by all



IAMs. The infilling process in AR6 was performed using an open-source Python software package called 'silicone' (Lamboll
et al., 2020).

Different infilling methods result in different levels of proportionality, consistency, and stability to small changes. In
AR6, the quantile rolling windows ("QRW") approach was chosen for the most reported emissions gases (aerosol precursor
emissions, volatile organic compounds and GHGs other than F-gases) because of the preference for high stability to small
changes in the database. This is a conservative approach that cannot result in infilled pathways being more extreme than the
database from which one infills ("infiller database"). To avoid artefacts for the QRW method with a biased emissions space
distribution in the infiller database, chlorinated and fluorinated gases are infilled based on a pathway with lowest root mean
squared difference ("RMS-closest"), ensuring a resulting emissions trend with consistency over time even when given few
input emissions scenarios. See **Table 1** for full details.

Where possible, missing emissions species are infilled from the harmonised AR6DB. Where the AR6DB does not
cover the emissions species, the CMIP6-emissions SSP dataset was used (**Table 1**).

Missing emission pathways from a scenario are infilled based on their relationship with $CO_2$-FFI. If $CO_2$-FFI is
strongly mitigated, the algorithm fills in pathways of other emission species from other scenarios in the AR6DB where $CO_2$-
FFI is mitigated similarly. This process is done based on emissions pathways that have already been harmonised. The AR6
WGIII report acknowledges that there is uncertainty in using this method, and therefore chose to only use the climate results
from scenarios where models natively provided at least $CO_2$-FFI, $CO_2$-AFOLU, $CH_4$, and $N_2O$. In principle, however, it would
be possible to produce a climate assessment for a scenario that only reports $CO_2$-FFI, but while this would increase model
diversity, such scenarios would still not be able to reflect the effect of policy choices that influence non-$CO_2$ emissions and
hence climate outcomes from sectors such as AFOLU, waste, and industrial use of $N_2O$ and F-gases.
**3.4 Climate emulators**
An extensive calibration and testing exercise of emulators to assess their suitability to reproduce assessed climate ranges has
been undertaken in AR6 WG1 and reported in the Cross-Chapter Box 7.1 of IPCC AR6 WGI (Forster et al., 2021; Smith et
al., 2021). The precedent for this exercise was given by the Reduced Complexity Model Intercomparison Project (RCMIP),
where Phase 2 of this project compared emulators' performances when constrained to hit pre-determined ranges of variables
including equilibrium climate sensitivity (ECS), transient climate response (TCR), observed global mean surface temperature,
ocean heat content change, transient climate response to cumulative emissions of carbon dioxide (TCRE), and radiative forcing
for species such as $CO_2$, $CH_4$ and aerosols (Nicholls et al., 2021a). One condition for an emulator to be used in the AR6 WGI
emulator analysis was that the emulator needs to comprise interactive carbon cycle and other gas cycle parameterisations so
that it can run from emission timeseries rather than from concentrations. In this exercise, emulators were driven by emission
timeseries of around 40 GHGs (with $CO_2$ broken down into $CO_2$-FFI and $CO_2$-AFOLU, components), short-lived climate
forcers, aerosol and ozone precursors, and external forcing from solar variability and volcanic stratospheric aerosol optical
depth. Four emulators contributed to the AR6 WGI exercise: MAGICCv7.5.3, FaIRv1.6.2, CICERO-SCM and OSCARv3.1.1.





While we look at annual mean temperatures, these emulators do not aim to capture any unforced internal variability of the
climate system.
MAGICCv7.5.3 and FaIRv1.6.2 were found to be able to reproduce Working Group I assessed climate variables to
within small error, with CICERO-SCM and OSCARv3.1.1 providing useful supporting information but with larger deviation
from the temperature changes as assessed by WGI. MAGICCV7.5.3, FaIRv1.6.2 and CICERO-SCM participated in the AR6
WGIII process and were included in the climate assessment workflow that provides 52 emissions species (see Table 1). The
connection of emulators to the workflow is done using the OpenSCM-runner interface (Nicholls et al., 2021b). Only
information of MAGICCv7.5.3 and FaIRv1.6.2 were used in the Summary for Policymakers, and in the results section of this
study we follow this focus on MAGICCv7.5.3 and FaIRv1.6.2, while we do some comparison with the climate outcomes of
CICERO-SCM. The scenario classification and reported medians are based on MAGICCv7.5.3, while reported ranges were
based on both MAGICCv7.5.3 and FaIRv1.6.2. As written in the WGI report, MAGICCv7.5.3 and FaIRv1.6.2 represent the
WGI assessment typically to within ±5% for central estimates of key climate change indicators, for instance for global warming
in 1995-2014 compared to 1850-1900, warming estimates along SSPs in the 21$^{st}$ century, current ERF compared to 1750 ERF
estimates, $CO_2$ airborne fractions under idealised experiments, and ocean heat content change between 1971 and 2018 (Forster
et al., 2021, Cross-Chapter Box 7.1, Table 2). For the upper and lower ranges, the difference with the WGI assessment is
within ±10% across more than 80% of metric ranges (Forster et al., 2021). Despite some identified limitations like the lack of
an interactive carbon cycle, and projecting lower warming than the best assessment along SSPs (e.g. -14% for SSP1-2.6 in
2081-2100 relative to 1995-2014), CICERO-SCM was assessed to represent historical warming very well, and can be used for
sensitivity analyses  (Forster et al., 2021).
**3.4.1 MAGICC**
MAGICC (Model for Assessment of Greenhouse gas Induced Climate Change) v7.5.3 is an emissions-driven Earth system
model emulator. Its atmosphere is represented as four interconnected boxes (northern and southern hemisphere ocean, northern
and southern hemisphere land). The ocean boxes are coupled to a 50-layer upwelling-diffusion-entrainment ocean model. A
full description of MAGICC can be found in Meinshausen et al. (2011b), with updates as described in Meinshausen et al.,
(2020) and Nicholls et al. (2021a). MAGICCv7.5.3 was calibrated using the Monte Carlo Markov Chain technique described
in Meinshausen et al. (2009), with an updated step to reweight the derived posterior to improve the match with the WGI
assessed ranges. The probabilistic distribution used in the climate assessment uses 600 ensemble members, balancing
computational costs with ensemble size. As also described in the documentation of the climate assessment workflow, the
MAGICCv7.5.3 binary and probabilistic distribution are not directly available within the climate assessment workflow but can
be accessed at https://magicc.org/download/magicc7 and used with the climate assessment workflow.





### 3.4.2 FaIR

FaIR (Finite-amplitude Impulse Response model) v1.6.2 is a fully open-source emissions-driven atmospheric model emulator with a state-dependent carbon cycle coupled to a two-ocean layer climate response module (Smith et al., 2018; Millar et al., 2017). The calibration for AR6 was performed using a 1-million-member prior ensemble. Parameters for the carbon cycle and climate response are derived from distributions based on CMIP6 models (Smith et al., 2018; Leach et al., 2021) and assessments made in AR6 WGI (Forster et al., 2021). This prior ensemble is simultaneously constrained on historical temperature (1850-2019), ocean heat content change (1971-2018), near-present-day (2014) $CO_2$ concentration, and airborne fraction of $CO_2$ in idealised 1% per year $CO_2$ increase experiments at the time of doubled $CO_2$, the latter of which is assessed by Chapter 5 of WGI (Canadell et al., 2021). Post-constraint checks are performed to ensure that ECS, TCR, and future warming lies close to the AR6 assessed ranges. The constrained ensemble used for probabilistic assessment contains 2,237 ensemble members. The calibrated, constrained ensemble for running FaIR is available as a JSON file from https://doi.org/10.5281/zenodo.5513022 (Smith, 2021a).

### 3.4.3 CICERO-SCM

The CICERO simple climate model (CICERO-SCM, Skeie et al., 2017) is also an emission-driven climate model emulator. The emulator consists of a carbon cycle model (Joos et al., 1996), simplified expressions relating emissions of components to forcing, either directly or via concentrations (Etminan et al., 2016; Skeie et al., 2017) and an energy balance/upwelling diffusion model (Schlesinger and Jiang, 1990; Schlesinger et al., 1992). The ensemble was based on a previously calibrated 30,400-member ensemble (Skeie et al., 2018). A 600-member subset of this ensemble was chosen to best fit the assessment made in WGI (Smith et al., 2021), with a technique also described in Nicholls et al., (2021a). For AR6 the ensemble was calibrated to the current temperature change from 1850-1900 to 1995-2014, with additional cut-offs for unrealistically low aerosol forcing or ECS values. The constrained ensemble for the climate assessment contains 600 members and a fully determined JSON file is available with the climate assessment workflow code (Kikstra et al., 2022a).

## 3.5 Climate categorisation of scenarios

### 3.5.1 Scenario classification used in AR6

The extensive climate assessment process provides increased confidence compared to previous assessments in the relationship between probabilistic temperature outcomes and the original modelled scenario. Therefore, the AR6 assessment used, like in SR1.5, a temperature-based set of classification rules, which are shown in **Table 2**. These categorisation criteria and their associated likelihoods are always associated with limits to global warming, looking at the simulated peak warming in the 21[st] century and the global mean surface temperature in 2100. For the categories that limit the global median temperature increase to less than 2°C above 1850-1900 levels (C1-C4), the categorization rules follow the same scheme as in SR1.5. Beyond these,





AR6 WGIII includes categories above relevant for higher emissions scenarios that cover the 2-2.5°C (C5), 2.5-3°C (C6), 3-
4°C (C7) and 4°C and higher (C8) global warming ranges, looking at modelled pathways until 2100. As already noted in
SR1.5, temperature-based categorisation is affected by uncertainty in future warming, uncertainty in past warming and the
reference period against which temperature levels are compared to (e.g., whether 'pre-industrial', which has a variety of
interpretations, or specifically 1850-1900 is taken as a reference period), but the relative difference between warming levels
and thus between temperature categories is more robust (IPCC, 2018).

### 3.5.2 Overshoot Degree Years

The categories C1 ("limit warming to 1.5°C (>50%) with no or limited overshoot") and C2 ("return warming to 1.5°C

(>50%) after a high overshoot") are separated based on their level of overshoot of 1.5°C. This separation in the classification
used in the IPCC report is purely based on the probability of overshoot (IPCC, 2022a), regardless of its magnitude or duration.
In practice, however, the separation based on probability also corresponds to the peak temperature of overshoot. Here, we
characterise this difference in overshoot for scenarios in more detail.

The extent and duration of the overshoot and the rate of change in overshoot temperatures are important for climate

impacts (Hoegh-Guldberg et al., 2018). Temperature levels may be largely independent of the path dependence of $CO_2$
emissions and removals (Tokarska et al., 2019) under limited overshoot with limited permafrost feedbacks (Gasser et al.,
2018), but many climate impacts are not (Seneviratne et al., 2018; Hoegh-Guldberg et al., 2018), including sea-level rise and
species extinction (IPCC, 2022b). For some impacts, the peak temperature during overshoot may be the deciding factor,
whereas in others it is more the integral of overshoot (i.e., the amount combined with the duration of overshoot), such as sea-
level rise in 2300 (Mengel et al., 2018).

To further analyse the characteristics of scenario categories beyond the analysis in AR6 we use the concept of

overshoot degree years (ODY), which is similar to what was shown as "overshoot severity" in Table 2.SM.12 in SR1.5 (Forster
et al., 2018), and was included in the metadata of the SR1.5 scenario database (Rogelj et al., 2018; Huppmann et al., 2018b)
as "exceedance severity". Inspired by Geden and Löschel, (2017) and recent scenario studies investigating temperature
overshoot (Drouet et al., 2021; Riahi et al., 2021; Johansson, 2021; Tachiiri et al., 2019).

In this study, we look at $ODY_{1.5}$ (in °C·years) as the cumulative overshoot degree-years above 1.5°C above 1850-

1900 from the start of each scenario until 2100, or the year specified otherwise: $\sum_t \max(0, T_t - T_\theta)$, where T is the annual
mean climatic global warming above 1850-1900, t is the year, and $\theta$ is the overshoot threshold. The indicator could allow for
defining limits for overshoot targets and thus be related to net-negative emissions in scenarios that return to below 1.5°C.
Additionally, it could be useful in studies that investigate the irreversibility of certain climate change impacts and could be an
indicator of the resilience of a system. For instance, in the case that some human system or ecosystem is unable to adapt
permanently but would be able to withstand up to 10 $ODY_{1.5}$ either through limited resilience or by using temporary adaptation
measures, this would indicate by when, under a certain scenario, the system may collapse. The AR6 Working Group II (WGII)
report on *Impacts, Adaptation and Vulnerability* (IPCC, 2022b) states with medium confidence that shorter duration and lower





levels of overshoot are projected to come with less severe impacts. ODY is not an indicator that can be used for all purposes,
as for some questions the rate of temperature change, or the level of peak warming reached in a given scenario may be more
relevant. Still, at the very least an indicator like this acknowledges that not only magnitude but also timescales are important
when assessing overshoot risks (Ritchie et al., 2021) and bridges the gap between stylized overshoot scenarios (Huntingford
et al., 2017). Analysing IAM scenarios in this way could be a useful link to the broader tipping points literature (Lenton et al.,
2019), and potentially inform climate change policy, impact, and adaptation studies.

### 3.5.3 Alternative policy-relevant scenario classifications

There are multiple possible indicators that can be chosen to classify and group scenarios (see the discussion above and e.g.,
Table 3.4 in AR4 WGIII (IPCC, 2007)). AR4 discussed this mainly as a matter of stabilisation of greenhouse gas concentrations
using a specific indicator as proxy along the chain from mitigation costs, through emissions to impacts. In response to the
introduction of temperature goals in international policy decisions and the spearheading of a temperature-aligned approach in
science-policy reports by the UN Environment Programme (Hare et al., 2010; Rogelj and Shukla, 2012; Rogelj et al., 2011),
SR1.5 and AR6 WGIII based their classifications on global warming levels. Global warming levels were used as one of the
integrating dimensions across the AR6 WGI report (Chen et al., 2021) and in the AR6 Working Group II (WGII) report on
*Impacts, Adaptation and Vulnerability*, as well as across WGs. However, it is also possible to append such a classification with
a mix of indicators, for instance to reflect a global climate agreement like the Paris Agreement. For example, the IPCC WGIII
AR6 report also reports a sub-category C1a of C1 scenarios (IPCC, 2022d). The additional criterion for this sub-category is
that net-zero GHG emissions are attained, generally in the second half of this century, which can be interpreted to reflect
Article 4.1 of the Paris Agreement (Fuglestvedt et al., 2018; Rogelj et al., 2021). Related examples of such mixed classifications
exist in the literature. For example, one recent paper proposes a specific interpretation of the Paris Agreement (Schleussner et
al., 2022), proposing that pathways can be seen as "Paris-compatible" if they (a) "[do] not ever have a greater than 66%
probability to overshoot 1.5°C", (b) "[are] *very likely* (90% chance or more) … not ever exceeding 2°C", and (c) achieve net-
zero greenhouse gas emissions using global warming potentials with a 100 year time horizon (GWP100).

### 3.6 Evaluating the effects of each step of the climate assessment workflow

The approach to emissions processing in AR6 WGIII was based on a combination of previous literature (Lamboll et
al., 2020; Gidden et al., 2018) and expert evaluation of the submitted pathways. The objective of this approach is to obtain an
unbiased, comparable, and plausible set of climate outcomes, in which each climate timeseries outcome reflects the original
pathway as truthfully as possible. To facilitate expanding and improving the methods, it is worth evaluating the appropriateness
of the set of tools in a quantitative manner. In this work, we provide an initial analysis by showing the effect on the total Kyoto
gases using a CO2-equivalent emission indicator (based on GWP100), for both harmonisation and infilling for each category.



## 4 Results

### 4.1 Characteristics of the full database.

The 1202 scenarios for which a climate assessment is available in AR6DB span a wide range of emissions pathways (**Figure 2A**). AR6 WGIII, including Chapter 3, used MAGICC for characterising the median estimates of global warming projections. The three climate emulators CICERO-SCM, FaIR, and MAGICC translate the set of infilled pathways in similar ways for atmospheric concentrations, with most distinctive differences for $N_2O$ (**Figure 2B**). Global mean surface temperatures above 1850-1900 levels are relatively similar between MAGICC and FaIR, while CICERO is colder (**Figure 2C**). Global-mean surface temperature change in IPCC WGIII AR6 (and here) is defined as degrees Celsius above the 1850-1900 mean, normalised to the best estimate of 0.85°C global warming for the period 1995-2014, as given by AR6 WGI.

In this manuscript, we focus on the median simulated climate outcomes of each scenario, with percentiles generally indicating percentiles over the selected scenario set. However, each climate variable, also including variables not discussed in this article such as ERF, ocean heat uptake, and $CO_2$ and $CH_4$ fluxes, as well as non-$CO_2$ warming for MAGICCv7.5.3, is available for each scenario for percentiles 5, 10, 16.7, 25, 33, 50, 67, 75, 83.3, 90, and 95 (Byers et al., 2022). The full AR6DB thus enables rich future studies of the uncertainty in multiple climate indicators for a large scenario set.

The database has scenarios (across all categories C1 to C8) with a very wide range for 2100 temperature outcomes, with its 5th to 95th percentile range stretching from 0.9-1.3°C to 3.2-3.8°C across scenarios, with the range for both the 5th and 95th percentiles arising from the differences across the three climate emulators. In 2050, the temperature outcome range is much smaller, covering a range of 1.4-1.6°C to 2.0-2.2°C above 1850-1900 (**Table 3**). The database thus covers a very broad spectrum of scenarios, going from groups of scenarios that reduce emissions fast enough to let temperatures decline in the second half of the century to scenarios that project increasingly fast warming. Still, it is noteworthy that on the one hand scenarios reaching 4°C warming this century reflect less than 5% of the scenarios in the AR6DB, and only very few scenarios in the database stay below 1.5°C by mid-century (except for CICERO, which is cooler and features a larger set of scenarios staying below 1.5°C, and was used only as a sensitivity case in the AR6 WGIII full report but was not included in the summary of results reported in the Summary for Policymakers).

### 4.2 Differences in climate emulators

The temperature classification in IPCC AR6 WGIII was done based on MAGICC. In high emissions scenarios MAGICC generally projects higher median outcomes than the other two emulators for the same set of scenarios (**Figure 3A**). The CICERO AR6-calibrated version projects the lowest amount of warming of the three emulators for all scenario categories.

For the two scenario categories with the most stringent temperature limits (C1 and C2), the medians of MAGICC and FaIR in 2100 are very close to each other. However, for these two categories MAGICC projects faster near-term warming than FaIR for the same emissions and thus MAGICC projects higher peak temperatures. Together, this implies a more negative zero emissions commitment (ZEC) in MAGICC compared to FaIR.



539   One way to investigate the difference in climate emulators is to look at the same scenario set and compare the relative
540 contributions of different emissions species to warming using median ERF. Looking at the ERF across scenarios for the
541 AR6DB split up in lower (C1-C4) and higher (C5-C8) temperature categories, it is clear that MAGICC and FaIR perform very
542 similarly, with slightly stronger negative aerosols forcing in MAGICC, and slightly stronger positive $CO_2$ forcing in FaIR
543 (**Figure 4A**). CICERO shows clearly lower $CO_2$ forcing than the other two emulators, while also having weaker negative
544 aerosols forcing.

545   Looking not at the ranges across scenarios, but rather at the climate uncertainties for each scenario in 2030, we see
546 that also the uncertainty ranges projected by FaIR and MAGICC are similar, though MAGICC projects somewhat higher
547 uncertainty ranges on near-term forcing from F-gases and aerosols (**Figure 4B**). CICERO does not have an interactive carbon
548 cycle representation and only represents uncertainties in aerosols, which are much smaller than in MAGICC and FaIR, where
549 uncertainty in aerosol-related ERF is especially large.

550 **4.3 Characteristics of scenario categories**

551   A multi-emulator comparison reveals that the temperature categorisation of a specific scenario can be quite sensitive
552 to small differences in how emissions are translated to global warming (**Figure 3B**). This is especially the case for the C1 and
553 C2 categories, with many scenarios in the AR6DB aiming at 1.5°C targets while warming is already 1.1°C for the period of
554 2011-2020 over 1850-1900 (IPCC, 2021). FaIR and MAGICC were assessed to cover the AR6 WGI assessment and its
555 uncertainties very well, which can be interpreted as generally approximating best estimate warming with an error up to 0.1°C
556 difference. While small in the broader context of uncertainty in the physical climate system, a 0.1°C difference in projected
557 peak temperature covers a non-trivial part of the difference between C1 and C2. Since FaIR projects slightly lower peak
558 temperatures than MAGICC, the number of scenarios classified in the AR6 temperature category C1 would double if the
559 classification would be repeated using FaIR. However, the number of scenarios in the wider set of 1.5°C and 2°C consistent
560 categories (C1-C4) is much more similar, with 758 for FaIR versus 687 for MAGICC.

561   In the supplementary material, we perform sensitivity experiments to explore the sensitivity to changes in absolute
562 warming level estimates of the number of scenarios within temperature categories C1-C3 (**Supplementary Figure 1**). Such
563 changes could happen for instance due to a change in the best estimate of historical warming since 1850-1900, an update of
564 the best estimate of $CO_2$ or aerosols forcing, or even due to choosing different harmonisation and infilling methods. If the peak
565 temperature estimates of all scenarios would have been 0.1°C higher, virtually no scenarios would be categorised as C1, while
566 the number would roughly double if peak temperature level estimates would be about 0.1°C lower. Furthermore, small
567 variations in the scenarios included in a category have a strong impact on the median net-zero GHG timing in C1, while the
568 effects on net-zero $CO_2$ in all categories and on net-zero GHG in C2 and C3 are less sensitive.





**4.4 Temperature overshoot**

Almost all scenarios are projected by MAGICC to overshoot 1.5°C, even in C1, with C3-C8 median warming estimates never returning to below 1.5°C this century (**Figure 5A-D**). The duration of overshoot in most C1 scenarios is limited to a few decades, generally starting in the 2030s, while some C2 scenarios are projected to have global warming of more than 1.5°C for most of the century (**Figure 5B-C**). The peak of overshoot in C1 scenarios is generally limited to up to 0.1°C, while scenarios in C2 are generally in the 0.1-0.4°C range. Hence even though categories C1 and C2 are defined solely based on their probability of exceeding 1.5°C, these scenarios are also practically distinguished by the amount by which they overshoot 1.5°C, which may be more relevant for climate change impact, vulnerability, and adaptation studies.

Using $ODY_{1.5}$ until 2100, we see that the severity of temperature exceedance above 1.5°C is also clearly differentiated by category, with different rates of increase of cumulative exceedance of 1.5°C after 2030 (**Figure 5E-F**). For instance, using the median of temperature estimates from MAGICC, we find that about three quarters of the scenarios in C1 stay below 2 $ODY_{1.5}$, and the 95th percentile across scenarios is slightly below 3 $ODY_{1.5}$ (**Figure 5E**). For more than three quarters of C1 scenarios, MAGICC projects a smaller than 33% probability (at the 67th percentile of warming) that overshoot severity would be more than 10 $ODY_{1.5}$. C4 scenarios are more likely than not below 2°C but do not return back to below 1.5°C. Their median ODY therefore steadily grows to over 20 $ODY_{1.5}$ by end-of-century for more than half of the scenarios. For more than half of the scenarios in C4 more than 10 $ODY_{1.5}$ by 2100 is projected with at least 67% chance, and about 33% chance that it would be more than 30 $ODY_{1.5}$. In higher temperature categories, $ODY_{1.5}$ increases ever-faster over time because temperatures keep increasing, resulting in median values of about 50 and 100 $ODY_{1.5}$ in 2100 for C6 and C8 in 2100 (**Figure 5F**).

**4.5 "Paris-compatible" scenarios using FaIR and MAGICC**

Using FaIR, 89 scenarios in the AR6DB would meet the three criteria for Paris-compatibility from Schleussner et al., (2022). Using MAGICC, 29 scenarios meet these criteria (**Figure 6A**). Net-zero $CO_2$ in the MAGICC scenario subset is reached around 2050, and before 2060 in the FaIR subset, looking at the interquartile range, with the median of both subsets being close to 2050. Net-zero GHG timing has a wider range across scenarios, with the medians across scenario subsets being about 15 years later (**Figure 6B**). The IPCC C1 category has a much wider range for GHG net-zero timing, with a few scenarios that do not have net negative GHG emissions but do have projected warming of less than 1.5°C in 2100. For net-zero $CO_2$ timing, the difference is small. The interquartile ranges for cumulative $CO_2$ emissions until net-zero $CO_2$ are 520-680Gt$CO_2$ for FaIR and 480-560Gt$CO_2$ for MAGICC. How remaining carbon budgets relate to temperature outcomes is strongly dependent on the level of non-$CO_2$ mitigation (Canadell et al., 2021; Riahi et al., 2022; IPCC, 2022a). However, even with the strongest non-$CO_2$ mitigation, no scenario with more than 1000Gt$CO_2$ cumulative emissions before reaching net-zero is deemed Paris-compatible according to these criteria using FaIR, or no more than 800Gt$CO_2$ using MAGICC.

The main climate difference between the "Paris-compatible" scenarios and the full C1 category is the amount by which temperature declines after its peak at 1.5-1.6°C in 2035-2055 (**Figure 6E**). For more than half of the scenarios in the





sub-group of 29 scenarios the temperature decline after 2040 is 0.3-0.4°C until 2100, whereas more than half of the other C1
scenarios see less than 0.2°C temperature decline post-2040 in this century (**Figure 6F**). The temperature decline in the "Paris-
compatible" (~0.06°C/decade) subset is about 2 times faster than the C1 subset that is not "Paris-compatible" (~0.03°C/decade,
**Figure 6G**). Such lower temperatures, which are also implied to decline beyond 2100 if no abrupt changes in emissions levels
and trends are assumed, come with lower risks related to for instance sea level rise and stresses related to heat extremes and
drought, given that temperatures would return towards current levels during the 22nd century. Conversely, the scenarios that
are in C1 but not classified as "Paris-compatible" are characterised by stronger $CO_2$ reductions by 2030 than the "Paris-
compatible" set. Those scenarios thus project rapid near-term reduction to avoid the large-scale net-negative $CO_2$ emissions
present in the second half of the century in scenarios that reach net-zero GHG emissions, as illustrated by **Figure 6 panel D**.

### 4.6 The effects of emissions processing in the AR6 workflow

The effects of harmonisation and infilling on input emissions pathways is small, when taken over the entire scenario database,
looking at GHGs for Kyoto Gases using GWP100 to calculate $CO_2$-equivalent values for $N_2O$, $CH_4$, and F-gases. The median
effect of harmonisation and infilling over the full scenario database is about $1GtCO_2$-eq/yr upwards in 2015, trending down to
zero towards the end of the scenario in 2100 (**Figure 7A**). However, some scenarios are affected by these processing steps
much more than others, with the 5th to 95th percentile range of about -2 to $4GtCO_2$-eq/yr in 2020 to -1 to $4GtCO_2$-eq/yr in 2100.
Investigating in which scenarios such changes occur, and for which emissions species, helps understanding differences with
other harmonization and infilling methods as discussed in the next section.
While the harmonisation effect decreases over time, the upper bound does not change much because it is dominated
by infilling effects in the second half of the century. Such a high infilling is almost always the result of high emissions scenarios
lacking detail in reporting F-gases, which can grow to about 7 $GtCO_2$-eq/yr in 2100 under high emissions growth (**Figure 7D**,
and category C7 in **Figure 7F**). As shown in **Figure 7B-D**, about half of the total effect on the outer ranges is due to the
harmonisation of $CO_2$-AFOLU, for which a large model spread exists, much in line with the uncertainty in historical databases
(Dhakal et al., 2022). For methane, and for all other long-lived greenhouse gases combined ($N_2O$ and F-gases), the median of
harmonisation is slightly positive. Most scenarios require little to no infilling for Kyoto GHGs measured in $CO_2$-equivalence,
but that does not mean that they are unaffected by infilling as they may still need significant infilling for aerosols and precursor
emissions. We do not find evidence that harmonisation and infilling introduce any particular strong bias across the climate
categories used in the IPCC AR6 WGIII report (**Figure 7E-F**). For harmonisation, for each category except C8 (which is the
smallest in terms of scenario numbers), the zero line falls well within the interquartile range, with the C2 median being most
negative, and the C4 median being the most positive (**Figure 7E**). In terms of infilling, only the C3 and C7 median effect
across scenarios show values larger than 0.3GtCO2-eq/yr due to infilling before 2040 (**Figure 7F**).
The total cumulative effect of infilling and harmonisation for the 2020-2100 period is small too (**Figure 7G and**
**Figure 8**). More than half of the scenarios in the AR6DB (730) have higher cumulative Kyoto gases emissions until 2100 after





harmonization and infilling, and 472 scenarios are lower, indicating that the infilling effect is not dominating the harmonisation effect. In part, the infilling effect is offset due to a large number of scenarios which report $CO_2$-AFOLU emissions levels higher than the 3.5$GtCO_2$/yr harmonised value in 2015, in combination with the late convergence target year for $CO_2$-AFOLU. Virtually all scenarios fall well within the +-500$GtCO_2$-equivalent band (**Figure 8B**). All except 1 of the C1-C5 scenarios fall well within the +-250$GtCO_2$-equivalent band (**Figure 8A**). Thus, this analysis does not show a clear pattern or bias pushing emissions up or down across categories. Rather, the harmonisation and infilling effect is mostly model-dependent, and the distribution of scenarios from certain IAM frameworks is not constant across temperature categories (**Supplementary Table 2**).

## 4.7 Changes in methods between SR1.5 and AR6 WGIII and their implications

The most recent and most rigorous scenario assessment until AR6 was done in SR1.5. Insights from IAM-based assessment have influenced the global science-policy discourse (van Beek et al., 2020, 2022) and are even referred to in outcomes from informed ambitions in the Glasgow Climate Pact (UNFCCC, 2021). The results of SR1.5 have been influential in the academic literature, influenced public debate around the world, and legitimised as well as challenged climate policy (Hermansen et al., 2021; Livingston and Rummukainen, 2020). It is thus crucial to understand how the AR6 assessment methods differ from the methods applied in SR1.5. Here we provide additional insights to Annex III.II.3.2.1. "Climate classification of global pathways" of AR6 WGIII (IPCC, 2022a). The analysis performed allows for isolating the approximate differences between SR1.5 and AR6 WGIII pertaining to each of the separate methodological steps of the climate assessment workflow, namely harmonisation, infilling, and climate emulation. The same set of emissions scenarios was run with five different configurations that are summarised in Table 4.

Analysing the scenarios available both in the AR6 database as well as in the SR1.5 database (see also IPCC (2022a)), using the climate emulator MAGICC, shows the effect that is due to partly compounding, partly offsetting changes in each stage of the climate assessment (**Figure 9A** and **Figure 9B**).

The effect of the climate emulator update and recalibration (MAGICC6 in SR1.5 versus MAGICCv7.5.3 in AR6 WGIII) means a slightly higher peak temperature for near-term temperature peaks (in C1 and C2), and a lower 2100 temperature for all scenario categories in AR6. The lower warming in 2100 in AR6 is more in line with the best estimate based on multiple lines of evidence in AR6 WGI, as expressed by a lower transient climate response in MAGICCv7.5.3 (for more, see Nicholls et al., in preparation).

The median harmonisation effect for C1 and C3 results in about 0.05°C lower temperature in the AR6 method, which may in part be explained by the difference in harmonisation year (2010 in SR1.5 versus 2015 in AR6 WGIII), as well as a later chosen convergence date for $CO_2$-AFOLU. However, our methods do not allow for an explicit analysis of these separate factors, which is beyond the scope of this paper.




The change in infilling methods results in slightly lower 2100 temperatures in AR6 for C1, but virtually zero for C3, and positive for high warming categories (particularly C7 and C8). This is not surprising because in SR1.5 infilling was done using RCP2.6, which is roughly consistent with C3. Scenarios in C1 see stronger mitigation, and thus the infilling method applied in AR6 WGIII also sees more strongly declining emissions from other GHGs that are being infilled.

Overall, the effect of updating climate assessment methods is typically less than 0.2°C, and for most scenarios less than 0.1°C (**Figure 9A**). This difference is small but non-negligible compared to precision of the climate emulators. The effect of the change in methods is always less than 25% of the projected warming in for that scenario, and typically less than 10% for both peak temperatures and 2100 temperatures (**Figure 9B**). Only for the C1 category is the change in 2100 a more substantial change in temperature when expressed as a percentage; this is due to the very small warming that occurs overall in this category, so that even small changes result in a more substantive percentage change of about 30% in the median. This, however, still corresponds to only an absolute median temperature difference of about 0.1°C.

There are a few outlier scenarios in C1 and C2, where the relative effect on projected warming in 2100 relative to 1995-2014 is more than 50%. These differences, both when negative and positive (up to ±0.2°C change) are mostly caused by a different infilling effect for scenarios that have a low projected warming until 2100, sometimes combined with a slightly more negative temperature drawdown after peak from the climate emulator. The effects are strongly scenario dependent. For instance, the change in 2100 projected temperature due to changes in infilling is opposite for AIM/CGE (AR6 infilling results in higher temperatures than SR1.5 infilling) and WITCH-GLOBIOM CD-LINKS_NPi2020_400 (AR6 infilling results in lower temperatures than SR1.5 infilling) scenarios.

Lastly, to understand the differences in reported summary characteristics across SR1.5 and AR6 WGIII, it is important to know the distributions of global warming that it is associated with. For instance, the scenarios in the lowest category in AR6 (C1) generally have higher peak and 2100 temperatures than the scenarios that featured in the analogous category in SR1.5 (**Figure 9C**). This reflects the continued growth seen in emissions in the past years, and therefore higher warming for the same (maximum feasible) rate of reductions in newer IAM scenarios published since SR1.5.




## 5 Discussion

### 5.1 Advancements in the AR6 report and where to go for AR7

The IPCC Sixth Assessment cycle saw important advancements in the climate assessment of the emissions scenario literature: from a concentration and forcing based approach in AR5 to a temperature based approach in SR1.5 and AR6 that more closely reflects policy needs; from the use of ad-hoc methods with important limitations for the completion and harmonisation of emissions in AR5 and SR1.5 to a carefully designed and more robust emissions scenario assessment across WGs in AR6; from the use of a single climate emulator in AR5 to the coordinated approach where WGI assessed and identified a set of emulators that most faithfully reflect the state-of-the-art understanding of global warming and its uncertainties. These have put the AR6 mitigation scenario assessment on a new level compared to earlier reports, but opportunities for further improvements in the next assessment cycle remain.

### 5.1.1 Moving beyond a binary quality vetting process

New methods could be devised to advance the methods used to vet scenarios that are considered. In the current AR6 process, a scenario was either found fit-for-purpose or not considered in the analysis. Future assessments could attempt to move beyond such a binary procedure and for example look at assigning relative weights to scenarios based on how well they match recent trends, and to increase the diversity of the evidence-base, with the global scenarios in the AR6DB being dominated by only a handful of modelling frameworks (**Supplementary Table 2**). In the report, it could lead to more information being available for partial assessments of scenarios. For the climate assessment, knowing which emissions trajectories are more in line with past trends could be used as information to determine how to infill a trajectory when it is missing. Moreover, new methods and evidence are required to assess the performance of emissions-driven climate emulators with higher confidence. Most of the CMIP exercises run concentration-driven experiments, instead of the emission-driven runs that would most directly inform emulator calibration and improvement. This research gap is particularly wide for understanding the climate consequences of scenarios with net negative $CO_2$ and GHG emissions.

### 5.1.2 Towards improving understanding of the role of aerosols in climate mitigation pathways

The role of aerosol and aerosol precursor emissions in warming projections of scenarios remains uncertain. This is in part due to large climate uncertainties that remain in the various aerosol-climate interactions and in emission inventories, and in part because of a lack of a broadly representative set of scenarios for regional aerosol emissions. There is also still a relatively modest focus of the IAM community on modelling alternative effects of aerosol and precursor emission processes, with aerosols generally not being part of scenario protocols in multi-model IAM studies.





### 5.1.3 Connecting to regional climate impact studies and IPCC WGII

The advancements in integration of insights and assessments from different research communities across climate mitigation and physical climate sciences in AR6 fell short of being fully reflected in the assessment of climate change impacts in WGII. However, the methods described in this paper could be one way to allow for such further integration. A closer connection between scenarios and the assessment of physical climate science on the one hand, and impact, vulnerability, and adaptation studies on the other hand could provide an extremely impactful contribution to the next IPCC assessment cycle. For instance, the current climate assessment workflow from emissions to global temperature change could be extended with another emulator towards regionally downscaled climate change projections, using tools such as MESMER (Beusch et al., 2020, 2022). A natural next step is to move one step further down the cause-effect chain from regional climate change to regional climate impacts. Using such a chain of emulators (Beusch et al., 2022) could enable probabilistic assessments of various types of impacts both at different global warming levels and under scenarios not considered by Earth System models, supplementing the evidence base used for adaptation and impact assessments made in IPCC WGII. Even without regional impacts, relevant global metrics can be obtained from this kind of workflow such as global sea-level rise. In turn, the scenario development and IAM community could draw lessons from such studies too, for instance by exploring parts of the impacts, vulnerability, and adaptation space that are found to be understudied.

### 5.2 Scenario classification approaches

In AR6 and multiple previous IPCC assessments, scenarios were grouped to enable describing the characteristics of a group of scenarios (e.g., emissions reductions) that have a similar relevant feature (e.g., change in global mean surface temperature). Future scenarios classifications can choose to review choices in two elements, namely (i) the chosen relevant feature and (ii) the tools used to evaluate how the chosen relevant feature relates to the scenario characteristic. When it comes to (i), one could for instance include other indicators beyond global temperature projections in the classification scheme when they are policy-relevant. This could include indicators on mitigation strategies, emissions trajectories, scenario and model design, other physical responses than global mean temperature, or climate impacts. In addition, the use of the median and $33^{rd}$, and $67^{th}$ percentiles of global mean surface temperature for the classification in AR6, as well as the chosen specific warming levels, should not be seen as set in stone. For instance, one could choose to set the upper bound for category C3 to <1.8°C at 50% probability, rather than <2.0°C at 67% probability. For (ii), AR6 WGIII used MAGICC to do the classification of scenarios. It would also be possible to use multiple climate emulators for classification, for instance by using a majority rule, a multi-model mean, or other ways of combining climate emulator distributions. In addition, the availability of information on multiple types of uncertainty (emissions, climate uncertainty within an emulator, multiple emulators) could be utilised to provide a confidence level of the assigned category classification.

Another aspect is the categorisation of scenarios, and the use of descriptive statistics. Describing larger scenario categories comes with further limitations, because summary statistics can conceal the underlying distribution or overemphasise





outliers. Further efforts could be made to describe key scenario characteristics by developing methods that correct for potential biases in the underlying scenario database, such as overrepresentation of scenarios from one specific modelling framework, or weightings based on feasibility, historical compatibility, or scenario similarity (Guivarch et al., 2022). Other topics that might be relevant for a more multi-dimensional categorisation could be a separation of scenarios by their temperature decline after their peak, or the associated reliance on net negative emissions to achieve this.

**5.3 Improving the understanding of the implications of overshoot**

Related to the question of impact is the question of overshoot. From Figure 5E-F we learn that each AR6 temperature category can be distinguished based on their $ODY_{1.5}$ timeseries, with almost all scenarios overshooting 1.5°C at least for a decade when using climate emulator MAGICC. Following the publication of the AR6 WGI, and much more strongly since the publication of AR6 WGII and WGIII, more focus has come on temperature overshoot. Many different peak-and-decline scenarios have been analysed in Chapter 3 of AR6 WGIII (Riahi et al., 2022), some with more pronounced overshoot than others. The discussion of overshoot in global climate policy is expected to be contentious due to its connection to the assumptions related to large-scale carbon dioxide removal or the potential that its presence in scenarios can delay strong mitigation policies while also potentially obscuring impact and feasibility risks of a temperature overshoot strategy (Maher and Symons, 2022; S.M. Smith, 2021b). While overshoot indicators like $ODY_{1.5}$ may immediately be a useful as an indicator to quantify differences in levels of overshoot between scenarios, further research is required to relate absolute levels of ODY to for instance climate impacts, loss and damage, and the risk of passing tipping points (Lenton et al., 2019) to be able to judge whether ODY or other temperature exceedance metrics could be a useful indicator to guide climate policies.

**5.4 Climate assessment workflow performance diagnostics and limitations, and further development**

In this manuscript, we have analysed the impact of changes in the climate assessment workflow between SR1.5 and AR6. The changes made between the two assessments drew on an expert judgement of the applicability of available methods, based on the available literature (Lamboll et al., 2020; Gidden et al., 2018, 2019), extensive knowledge of the AR6 scenario database, and experience from previous IPCC reports. To enable assessing the climate outcomes of different climate assessment workflow methods, and to help determine whether such a change in methods is an improvement, a more systematic analysis is required. Such a more systematic analysis could involve establishing a reference case, specify a set of "standard experiments" to be performed, and develop a set of diagnostics to evaluate the differences between method choices. In this manuscript, we have used GWP100 which is available in the AR6DB (Byers et al., 2022) to analyse the impact of the harmonisation and infilling of emissions trajectories. However, such an analysis is limited because it does not capture all climatically active species, like aerosols, and because GWP100 is only one out of multiple possible metrics. Alternative metric choices would not alter the climate outcome but could significantly affect the reported date at which net zero GHG emissions are reached (Dhakal et al., 2022; Figure 2 SM.10).



### 5.4.1 Improvements for harmonisation

This paper analysed the changes in temperature estimates as the result of different methods using an ad-hoc setup. This setup could serve as inspiration for a future diagnostic tool, and the development of benchmarks. Future work could consider extending or adjusting the decision tree currently available in 'aneris'. For instance, to facilitate earlier convergence times for instance for $CO_2$ emissions in scenarios that reach and sustain net-zero $CO_2$ emissions, the decision tree could incorporate the convergence year dependent on the scenario design. A significant limitation of the harmonisation part of the workflow comes from the uncertainty in historical emissions, and how this is projected forwards. Harmonisation collapses this uncertainty, sometimes updating emissions estimates that are out-of-date but other times forcing sets of estimates predicated on different measurements to agree with each other. In some cases, the trends of harmonised data can be markedly different to the trends in the original pathways - for instance, if historic emissions of an F-gas were overestimated but are predicted to fall sharply, the return to the original value can cause a net positive gradient. Going forwards, it would be good to investigate the impact of historic emissions choices and uncertainty on results.

### 5.4.2 Improvements for infilling

In a similar fashion, infilling performance can also be improved in a few different ways. One way would be to simply have a wider variety of modelled scenarios including especially aerosols and individual fluorinated gases, allowing for more differentiated infilled pathways. For some species however, such as aerosols and ozone precursors, more research is needed to confidently select the most reasonable pathways or to infill a trajectory when it is missing. Another more advanced way would be to consider assigning weights to emissions trajectories in the scenario database. Lastly, and perhaps most influentially, future workflows could consider developing an infilling method decision tree for each emissions species. In AR6, two different methods and infiller databases are used, but always with the same lead gas, $CO_2$ from energy and industrial processes. For example, it may be preferable to let black carbon act as lead component for filling in an organic carbon timeseries, when available.

### 5.4.3 The order of emissions processing steps

Another particular choice that could be evaluated in future work is the order of emissions processing. In AR6, following SR15, scenario vetting is done first, harmonisation second, and infilling (based on a harmonised set of emissions trajectories) last. Such a strategy ensures that the pathways that are infilled are always starting from a reasonable point and influenced less by differences in historical emissions databases. Moreover, in this way two pathways that are identical except for when they were last harmonised, should have the same infilled emissions. However, it would also be possible to do infilling before harmonisation, which would derive inter-species statistics used for infilling more directly from the modelled processes in the IAMs. This can only be guaranteed if they are infilled after harmonisation to the latest values. Lastly, by reducing the range of projections when using the QRW method, the risk of out-of-sample infilling is reduced.





### 5.4.4 Potential for further development of a community tool


The *climate-assessment* workflow is available as an installable open-source Python package with an MIT license (**Kikstra et**
**al.**). The code utilises functions of existing scientific software packages including 'pyam' (Huppmann et al., 2021) and has
been parallelised to enable doing runs of many scenarios. It could be used as a community tool for scenario assessment that
enables both easier access to well-calibrated climate emulators and the possibility to assess a wider range of scenarios due to
the possibility of infilling emissions trajectories. Such access to climate assessment tool can facilitate the development of
socioeconomic scenarios, for instance when new models only have the ability to model a limited number of emission species.
Results have already been used to allow for calculating the non-$CO_2$ contribution to warming which is used to estimate the
remaining carbon budget (Lamboll and Rogelj, 2022).

Crucially, through the 'openscm' interface (Nicholls et al., 2021b), it is possible to connect more climate emulators
to this workflow as well, which could enable easier intercomparisons in the future. Firstly, to enable a robust assessment of
climate mitigation pathways, a multi-emulator setup is crucial to understand both differences between the multiple models out
there, including those that participated in RCMIP (Nicholls et al., 2021a). Secondly, having a wider set of simple climate
models available and connected to this workflow could allow wider applications as the models differ in the detail and methods
with which processes are modelled, and thus also differ in what variables can be projected alongside scenarios.

### 6 Conclusions


The IPCC Sixth Assessment Report on the Mitigation of Climate Change (IPCC, 2022c) evaluated the climate outcomes of a
very broad range of scenarios. This manuscript further documents and evaluates the climate assessment workflow that allowed
for this analysis and has further explored elements related to compatibility with the Paris Agreement, temperature overshoot,
and the differences between climate emulators. The *climate-assessment* package introduced with this manuscript can serve as
a tool that currently can support modellers to project climate outcomes of scenarios with emissions information, even if only
several major emissions species were modelled. Future work could take this work as a start to further expand the coverage of
the causal chain from emissions to climate impacts, by extending the workflow beyond global climate characteristics toward
regional or local climate change projections of temperature and precipitation and calculated climate impacts.



## Acknowledgements

We would like to thank Katherine Calvin, Gaurav Ganti, Philip Hackstock, Felix Schenuit, and Jim Skea for their helpful suggestions and discussions on the development and diagnosis of the climate assessment workflow and this manuscript.

J.S.K. was supported by the UK Natural Environment Research Council under grant agreement NE/S007415/1. C.J.S. was supported by a NERC/IIASA Collaborative Research Fellowship (NE/T009381/1).

R. L., J.R. and P.M.F. received funding from the European Union's Horizon 2020 research and innovation programme under grant agreement No 820829 (CONSTRAIN).

G.P.P. was funded by the European Union's Horizon 2020 research and innovation programme under grant agreement No 821003 (4C) and No 820846 (Paris Reinforce).

J.S.K, E.B, K.R, R.S, E.K., D.P.V, acknowledge funding from the European Union's Horizon 2020 research and innovation programme under grant agreement No 821471 (ENGAGE). Funding from IIASA's National Member Organizations that has supported core activities including development of the Scenario Explorer and Database infrastructure is gratefully acknowledged.

K.R. acknowledges funding from the European Research Council (ERC) under the European Union's Horizon 2020 research and innovation programme (grant agreement No 951542) (GENIE).

Z.R.J.N., J.L., M.M., and J.R. have received funding from the European Union's Horizon 2020 research and innovation programme under grant agreement No 101003536 (ESM2025).

A.AK was supported by the Engineering and Physical Sciences Research Council, United Kingdom. Grant/Award Number: EP/P022820/1.

## Author contribution

J.S.K. wrote the first draft of the manuscript and produced the figures and tables; J.S.K. and Z.R.J.N. coordinated and developed the climate assessment workflow, with considerable help in coding from J.L., and additional work done by R.L., C.J.S., and M.S.; Z.R.J.N., J.L. and M.M. developed MAGICCV7.5.3 and produced the output for this climate emulator; C.J.S. and P.M.F. developed FaIRv1.6.2 and produced the output for this climate emulator; M.S., R.B.S, and B.H.S. developed CICERO-SCM and produced the output for this climate emulator; R.L. and J.R. developed *silicone* and R.L. implemented its methods in the climate assessment workflow, with support from Z.R.J.N., J.R., and J.S.K.; M.G. developed *aneris* and supported J.S.K. in implementing its methods in the climate assessment workflow; L.W. professionalised the codebase and supported the documentation of the climate assessment workflow; E.B. was responsible for maintaining and vetting the AR6DB, and calculating extensive metadata for the database, with considerable vetting analysis input from E.K, K.v.d.W. and J.S.K.; K.R. and R.S. coordinated the general use of the climate assessment workflow output and provided expert input on the methods applied during multiple assessment rounds in the IPCC process, in cooperation with E.K., G.P., D.P.v.V., P.M.F.,



M.M., J.S.F., J.R., A.A.K., A.R., J.S.K., E.B., and Z.R.J.N. who also facilitated the coordination and integration of information
between WGI and WGIII. All authors contributed to writing and reviewing the manuscript.

**Competing interests**

The authors declare that they have no conflict of interest.

**Code availability statement**

The 'climate-assessment' Python package is available on PyPi (https://pypi.org/project/climate-assessment), on GitHub
(https://github.com/iiasa/climate-assessment), and Zenodo (https://doi.org/10.5281/zenodo.6624519).
The code of the climate assessment workflow used version 0.1.0 of 'climate-assessment', available at
https://zenodo.org/record/6624520, or https://github.com/iiasa/climate-assessment/releases/tag/v0.1.0.
The full documentation of the AR6 version of the climate assessment package is available at https://climate-
assessment.readthedocs.io. The code includes a tutorial Jupyter notebook in which a simple climate assessment workflow run
with FaIR is performed.

**Data availability statement**

The scripts that were used to produce the figures and tables in the main text is available at Zenodo
https://zenodo.org/record/6610604 (Kikstra, 2022).
'aneris', 'silicone', and 'openscm-runner', are used directly in the AR6 workflow, with code available at
https://github.com/iiasa/aneris/releases/tag/v0.3.1, https://github.com/GranthamImperial/silicone/releases/tag/v1.2.1, and
https://github.com/openscm/openscm-runner/releases/tag/v0.9.1, respectively.
The used infiller database (version 1.0) is available separately at Zenodo https://zenodo.org/record/6390768 (Kikstra et al.,
2022b), while the historical emissions database (file "history_ar6.csv") is available with the *climate-assessment* repository as
documented at Zenodo (Kikstra et al., 2022a), and on GitHub (https://github.com/iiasa/climate-
assessment/blob/main/src/climate_assessment/harmonization/history_ar6.csv).
Emulators:
The CICERO-SCM model is available directly through the AR6 workflow, through the *openscm-runner* package.
The CICERO-SCM calibrated and constrained parameter set is made available with the *climate-assessment* package
at https://github.com/iiasa/climate-assessment/blob/main/data/cicero/subset_cscm_configfile.json, and on Zenodo
(file "subset_cscm_configfile.json", Kikstra et al., 2022a).



The FaIR model is available directly through the AR6 workflow, through the *openscm-runner* package, with code
available at: https://github.com/OMS-NetZero/FAIR/. The FaIRv1.6.2 calibrated and constrained parameter set is
available at https://doi.org/10.5281/zenodo.5513022 (Smith, 2021a), and download instructions are provided with the
*climate-assessment* package.
The   MAGICC   model   with   the   calibrated   and   constrained   parameter   is   available   at
https://magicc.org/download/magicc7, and once downloaded and installed can be used with the workflow.
The emissions data is available on the Downloads page of the AR6 Scenario Database hosted by IIASA:
https://data.ece.iiasa.ac.at/ar6, DOI: https://doi.org/10.5281/zenodo.5886912 (Byers et al., 2022).



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





| Emission species | Harmonisation Method | Reason for chosen method | Infilling Method | Infiller database |
|---|---|---|---|---|
| BC | Using default aneris decision tree | Default following Gidden et al. | QRW | AR6 database |
| CH₄ | Using default aneris decision tree | Default following Gidden et al. | -/QRW* | AR6 database |
| CO₂-AFOLU | Linearly reduce the difference between harmonized and non-harmonized with projected point of convergence in 2150. | High historical variance, but using offset method to prevent diff from increasing when going negative rapidly | -/QRW* | AR6 database |
| CO₂-FFI | Calculate the relative difference in 2015 and linearly reduce this ratio of the difference between harmonized and non-harmonized with projected point of convergence in 2080. | Default following Gidden et al, with ratio to have better performance for negative emissions pathways, 2080 instead of SR1.5 2050 because there is a wider set of scenarios covered, with many scenarios without strong mitigation in the database. | - | AR6 database |
| CO | Linearly reduce the difference between harmonized and non-harmonized with projected point of convergence in 2150. | High historical variance | QRW | AR6 database |
| N₂O | Using default aneris decision tree | Default following Gidden et al. | -/QRW* | AR6 database |
| NH₃ | Using default aneris decision tree | Default following Gidden et al. | QRW | AR6 database |
| NOₓ | Using default aneris decision tree | Default following Gidden et al. | QRW | AR6 database |
| OC | Linearly reduce the difference between harmonized and non-harmonized with projected point of convergence in 2150. | High historical variance | QRW | AR6 database |
| Sulfur | Using default aneris decision tree | Default following Gidden et al. | QRW | AR6 database |
| VOC | Linearly reduce the difference between harmonized and non-harmonized with projected point of convergence in 2150. | High historical variance | QRW | AR6 database |
| HFC134a | Keep the ratio of the difference between the harmonized and non-harmonized pathways constant over the full pathway. | Low model reporting confidence | RMS-closest | AR6 database |
| HFC143a | Keep the ratio of the difference between the harmonized and non-harmonized pathways constant over the full pathway. | Low model reporting confidence | RMS-closest | AR6 database |
| HFC227ea | Keep the ratio of the difference between the harmonized and non-harmonized pathways constant over the full pathway. | Low model reporting confidence | RMS-closest | AR6 database |





| | | | | |
|---|---|---|---|---|
| HFC23 | Keep the ratio of the difference between the harmonized and non-harmonized pathways constant over the full pathway. | Low model reporting confidence | RMS-closest | AR6 database |
| HFC32 | Keep the ratio of the difference between the harmonized and non-harmonized pathways constant over the full pathway. | Low model reporting confidence | RMS-closest | AR6 database |
| HFC43-10 | Keep the ratio of the difference between the harmonized and non-harmonized pathways constant over the full pathway. | Low model reporting confidence | RMS-closest | AR6 database |
| HFC125 | Keep the ratio of the difference between the harmonized and non-harmonized pathways constant over the full pathway. | Low model reporting confidence | RMS-closest | AR6 database |
| $SF_6$ | Keep the ratio of the difference between the harmonized and non-harmonized pathways constant over the full pathway. | Low model reporting confidence | RMS-closest | AR6 database |
| $CF_4$ (PFC) | Linearly reduce the difference between harmonized and non-harmonized with projected point of convergence in 2150. | High historical variance | RMS-closest | AR6 database |
| $C_2F_6$ (PFC) | Linearly reduce the difference between harmonized and non-harmonized with projected point of convergence in 2150. | High historical variance | RMS-closest | AR6 database |
| $C_6F_{14}$ (PFC) | Linearly reduce the difference between harmonized and non-harmonized with projected point of convergence in 2150. | High historical variance | RMS-closest | AR6 database |
| $CCl_4$ | - | - | RMS-closest | CMIP6-SSPs |
| CFC11 | - | - | RMS-closest | CMIP6-SSPs |
| CFC113 | - | - | RMS-closest | CMIP6-SSPs |
| CFC114 | - | - | RMS-closest | CMIP6-SSPs |
| CFC115 | - | - | RMS-closest | CMIP6-SSPs |
| CFC12 | - | - | RMS-closest | CMIP6-SSPs |





| | | | | |
|---|---|---|---|---|
| $CH_2Cl_2$ | - | - | RMS-closest | CMIP6-SSPs |
| $CH_3Br$ | - | - | RMS-closest | CMIP6-SSPs |
| $CH_3CCl_3$ | - | - | RMS-closest | CMIP6-SSPs |
| $CH_3Cl$ | - | - | RMS-closest | CMIP6-SSPs |
| $CHCl_3$ | - | - | RMS-closest | CMIP6-SSPs |
| HCFC141b | - | - | RMS-closest | CMIP6-SSPs |
| HCFC142b | - | - | RMS-closest | CMIP6-SSPs |
| HCFC22 | - | - | RMS-closest | CMIP6-SSPs |
| HFC152a | - | - | RMS-closest | CMIP6-SSPs |
| HFC236fa | - | - | RMS-closest | CMIP6-SSPs |
| HFC365mfc | - | - | RMS-closest | CMIP6-SSPs |
| Halon1202 | - | - | RMS-closest | CMIP6-SSPs |
| Halon1211 | - | - | RMS-closest | CMIP6-SSPs |
| Halon1301 | - | - | RMS-closest | CMIP6-SSPs |
| Halon2402 | - | - | RMS-closest | CMIP6-SSPs |
| $NF_3$ | - | - | RMS-closest | CMIP6-SSPs |
| $C_3F_8$ (PFC) | - | - | RMS-closest | CMIP6-SSPs |
| $C_4F_{10}$ (PFC) | - | - | RMS-closest | CMIP6-SSPs |



| | | | | |
|---|---|---|---|---|
| $C_5F_{12}$ (PFC) | - | - | RMS-closest | CMIP6-SSPs |
| $C_7F_{16}$ (PFC) | - | - | RMS-closest | CMIP6-SSPs |
| $C_8F_{18}$ (PFC) | - | - | RMS-closest | CMIP6-SSPs |
| $cC_4F_8$ (PFC) | - | - | RMS-closest | CMIP6-SSPs |
| $SO_2F_2$ | - | - | RMS-closest | CMIP6-SSPs |

**Table 1: Harmonisation and Infilling methods by emissions species as applied in AR6 WGIII. An asterisk (\*) means that the methods are in place, but not used in the report because of the 3-gas check. \*\*not harmonised because not available in used historical emissions database.**




| Description | Classification rules |
| --- | --- |
| C1: limit warming to 1.5°C (>50%) with no or limited overshoot | <1.5°C peak warming with ≥33% chance and <1.5°C end of century warming with >50% chance |
| C2: return warming to 1.5°C (>50%) after a high overshoot | <1.5°C peak warming with <33% chance and <1.5°C end of century warming with >50% chance |
| C3: limit warming to 2°C (>67%) | <2°C peak warming with >67% chance |
| C4: limit warming to 2°C (>50%) | <2°C peak warming with >50% chance |
| C5: limit warming to 2.5°C (>50%) | <2.5°C peak warming with >50% chance |
| C6: limit warming to 3°C (>50%) | <3°C peak warming with >50% chance |
| C7: limit warming to 4°C (>50%) | <4°C peak warming with >50% chance |
| C8: exceed 4°C warming (≥50%) | ≥4°C peak warming with ≥50% chance |


**Table 2: Temperature classification rules used in AR6 WGIII, where a scenario is placed in the lowest category where**
**it meets the classification rule.**





| | Climate emulator | 2050 | 2100 |
|---|---|---|---|
| **C1** | | | |
| | MAGICCv.7.5.3 | 1.5 (25)<br>1.6 (50)<br>1.6 (75) | 1.2 (25)<br>1.3 (50)<br>1.4 (75) |
| **C3** | | | |
| | MAGICCv.7.5.3 | 1.7 (25)<br>1.7 (50)<br>1.8 (75) | 1.6 (25)<br>1.6 (50)<br>1.7 (75) |
| **Full database** | | | |
| | MAGICCv.7.5.3 | 1.6 (5)<br>1.7 (25)<br>1.8 (50)<br>1.9 (75)<br>2.2 (95) | 1.3 (5)<br>1.6 (25)<br>1.8 (50)<br>2.5 (75)<br>3.8 (95) |
| | CICERO-SCM | 1.4 (5)<br>1.5 (25)<br>1.6 (50)<br>1.7 (75)<br>2.0 (95) | 0.9 (5)<br>1.2 (25)<br>1.4 (50)<br>1.9 (75)<br>3.2 (95) |
| | FaIRv1.6.2 | 1.5 (5)<br>1.6 (25)<br>1.7 (50)<br>1.8 (75)<br>2.1 (95) | 1.3 (5)<br>1.5 (25)<br>1.7 (50)<br>2.3 (75)<br>3.5 (95) |

**Table 3: Median global temperature statistics of the full scenario database, by climate model, with the percentiles over**
**the scenarios in parentheses for each row.**





| ID | Name | Harmonisation | Infilling | Climate emulator |
|---|---|---|---|---|
| (1) | SR1.5 report | SR1.5 | SR1.5 | MAGICC6 (AR5/SR1.5) |
| (2) | Climate emulator isolation | SR1.5 | SR1.5 | MAGICCV7.5.3 |
| (3) | Harmonisation algorithm on top of climate emulator isolation | AR6 algorithm, with scenarios harmonised in 2010 (rather than 2015 as is the default for the AR6 WGIII work) | SR1.5 | MAGICCV7.5.3 |
| (4) | AR6 workflow with 2010 harmonisation | AR6 algorithm, with scenarios harmonised in 2010 (rather than 2015 as is the default for the AR6 WGIII work) | AR6 | MAGICCV7.5.3 |
| (5) | AR6 workflow | AR6 | AR6 | MAGICCV7.5.3 |

| | Total | Harmonisation | Infilling | Climate emulator |
|---|---|---|---|---|
| **Calculating difference due to method change** | (5) – (1) | (3) - (2) + (5) - (4)<br><br>((3) - (2) is the change in algorithm, (5) - (4) is the change in harmonisation year) | (4) – (3) | (2) – (1) |

**Table 4: Summary of five climate assessment runs done to isolate the approximate changes in the temperature outcome attributable to each step of the climate assessment workflow.**



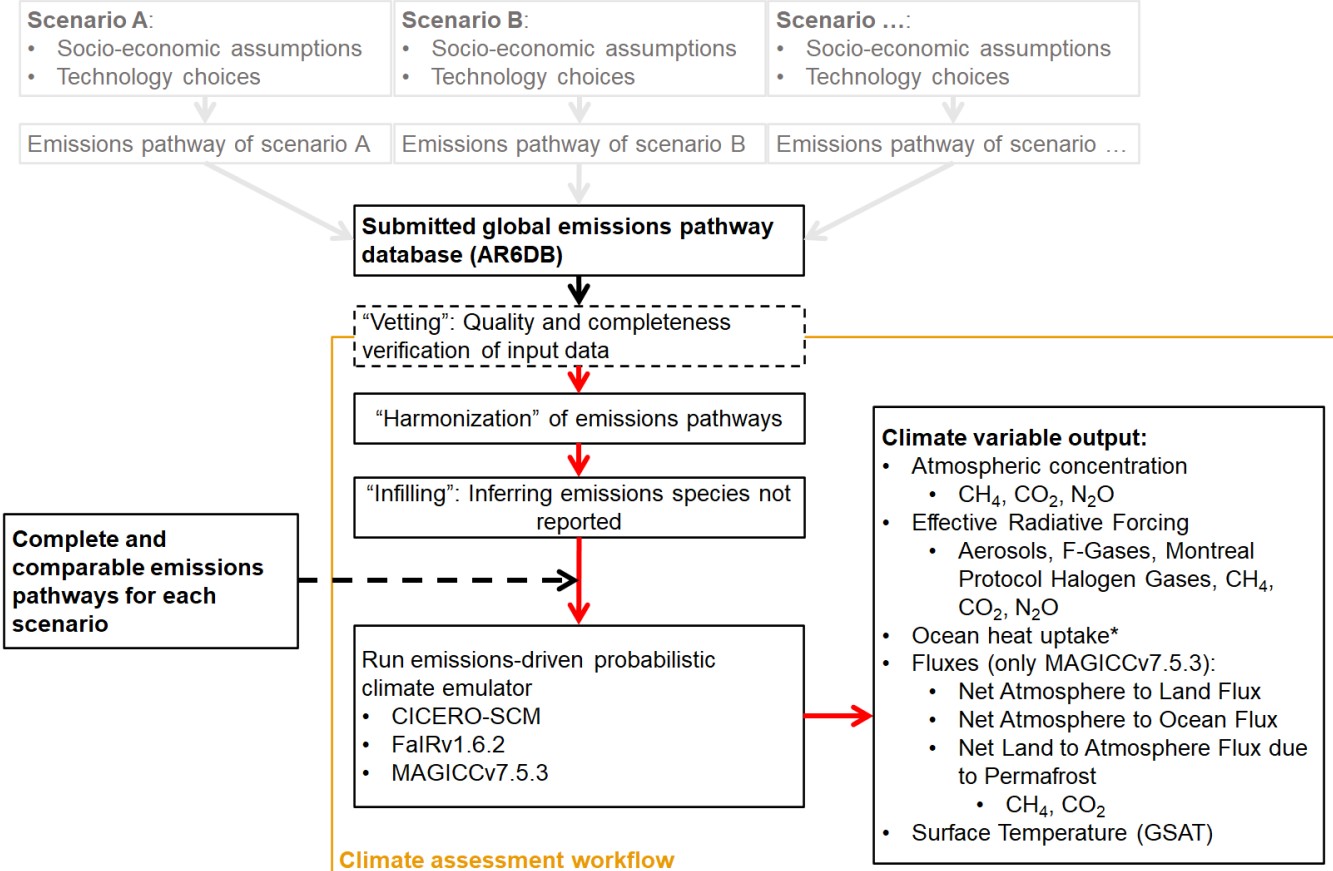

**Figure 1: The steps of the "climate assessment workflow". Overview of climate assessment processing steps applied in the Working Group III contribution to the IPCC Sixth Assessment Report. *Ocean heat uptake was provided by FaIRv.1.6.2 and MAGICCv7.5.3 in AR6.**





**Figure 2: Summary statistics (A: emissions, B: atmospheric concentrations, C: global mean surface temperature) over time across all scenarios in the AR6DB that received a temperature classification and across scenarios in AR6 temperature categories C1 and C3. Panel A shows emissions as modelled by IAMs (Native), after harmonisation (Harmonized), and after infilling missing reported emissions (Infilled). Panel B and panel C show climate outcomes per climate model, using the median value of each variable from the climate emulator probabilistic distributions.**



1280

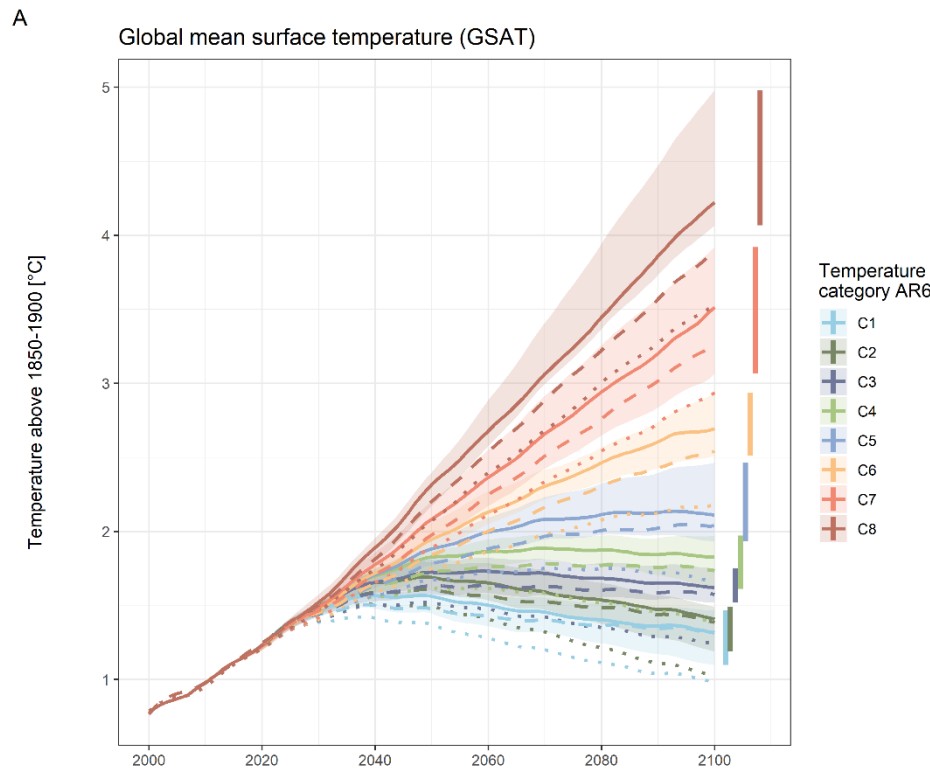

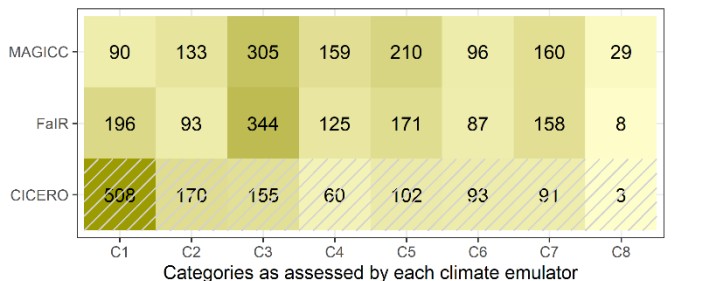

1281

**Figure 3: Median global surface temperatures above the mean of 1850-1900 as simulated for scenarios in the AR6DB, with scenarios grouped by the classification as in AR6WGIII. Medians are shown for all three uses climate emulators (CICERO-SCM: dotted, MAGICCv7.5.3: solid, and FaIRv1.6.2: dashed), while the 5th-95th percentile range is only shown for MAGICCv7.5.3. The number of scenarios classified in each group are shown in the bottom panel. CICERO-SCM numbers are hashed to indicate that AR6 WGI assessed especially the used parameterisations of MAGICCv7.5.3 and FaIRv1.6.2 to closely reflect the IPCC assessment, with CICERO-SCM for its AR6 calibration being used in WGIII only for sensitivity analysis around to capture climatic uncertainty ranges.**



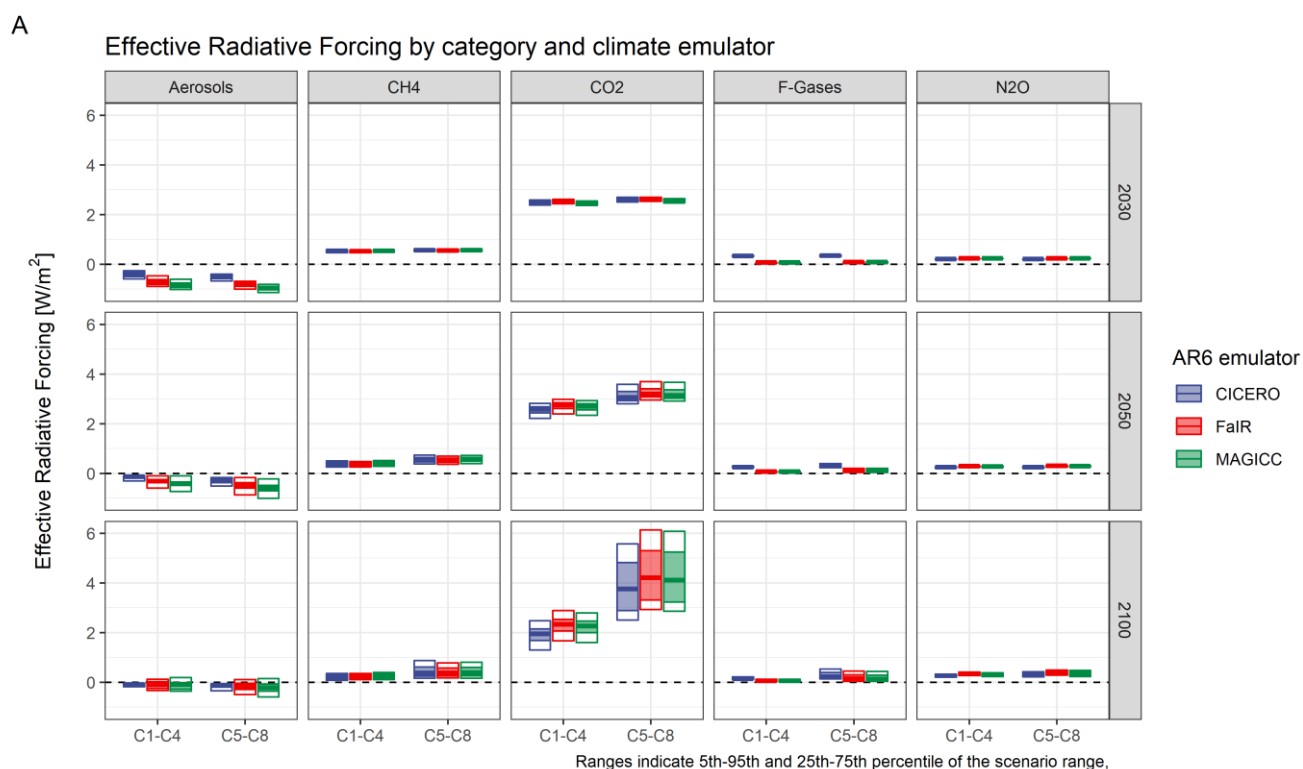

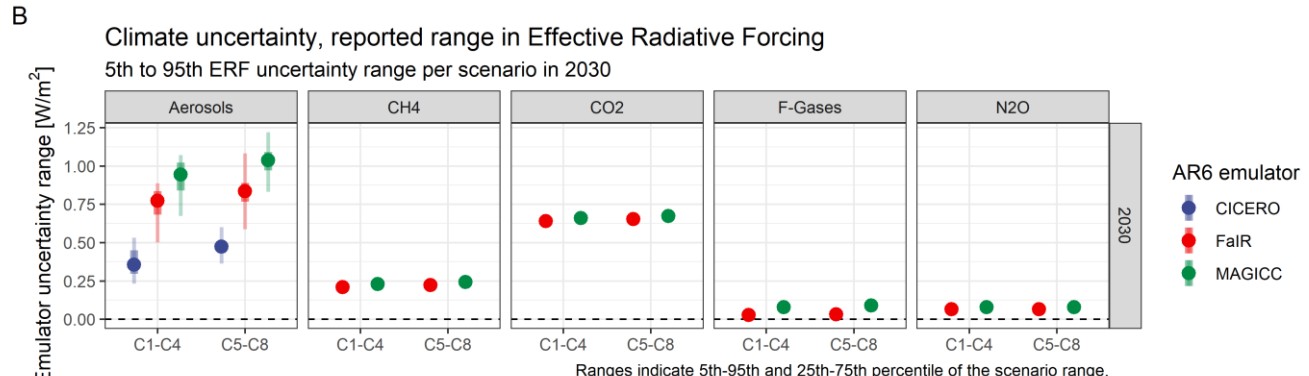

**Figure 4: Panel A: Effective Radiative Forcing (ERF) statistics across AR6 scenario database subsets as categorised by AR6WGIII**
**using MAGICC, for CO₂, CH₄, F-gases, and aerosols at different points in time for three climate emulators. Panel B: climate**
**uncertainty for every scenario as represented in projected ERF in 2030 for each climate emulator. For some gases this uncertainty**
**is relatively well-known (a known unknown) but for aerosols this uncertainty in forcing is still relatively unconstrained and depends**
**heavily on the magnitude and mix of emissions within a scenario (an unknown unknown).**







**Figure 5: The duration and magnitude of overshoot and exceedance of 1.5°C global warming above 1850-1900 for scenarios in the AR6 temperature categories. Panel A: projected median global mean surface temperature for scenarios in C1 and C2. Panel B-C: magnitude and duration of overshoot of 1.5°C in C1 and C2 scenario. Panel D: magnitude of 1.5°C exceedance of scenarios in C3-C8. Panel E: projected increase of ODY$_{1.5}$ over time for temperature categories C1-C4, at 33%, 50%, and 67% probability. Panel F: projected cumulative exceedance of 1.5°C expressed as ODY$_{1.5}$ in 2100 for temperature categories C1-C8, at 33%, 50%, and 67% probability.**




**Figure 6: Characteristics of "Paris-compatible" scenarios compared to the C1-C4 categories from IPCC AR6 WGIII using the FaIR and MAGICC emulators. 'Paris' here is short for "Paris-compatible" and uses the criteria from (Schleussner et al., 2022), being (a) "not ever have a greater than 66% probability to overshoot 1.5°C", (b) "*very likely* (90% chance or more) … not ever exceeding 2°C", and (c) achieving net-zero greenhouse gas emissions using global warming potentials over a 100 year period (GWP100).**





**Figure 7: Effect of harmonisation and infilling processing steps on Kyoto gases emissions trajectories. Panel A-D: the effect of harmonisation and infilling over time on GHGs, CO2 AFOLU, and CH4, for the full AR6DB. Panel E-F: effects of emissions processing by AR6 temperature category. Panel E: the effect on GHGs in 2015 due to harmonisation. Panel F: the effect of harmonisation and infilling on GHGs over time. Panel G: the cumulative effect of emissions processing until 2100 over the projected global warming.**





**Figure 8: Kyoto gases cumulative 2020-2100 for infilled and model reported by category. Each dot represents one long-term full century scenario. If model input would perfectly align with the used historical database and model all emissions species, or if harmonization and infilling cancel each other exactly the input GHG emissions would be the same as the GHG emissions after harmonisation and infilling. A spread on both sides of the line would be expected if historical emissions uncertainty would dominate and the use of different modelled historical emissions would not have a particular bias compared to the emissions estimate used for harmonisation. On the other hand, if many scenarios miss information on some important GHGs, dots would appear predominantly on the right of the line.**





**Figure 9: Differences in the AR6 and SR15 climate assessment workflow steps (panels A and B), and the temperature outcome distributions (panel C), using MAGICC. In Panel A and B, the AR6 temperature categories for a specific scenario were used. In Panel C, we use the categories as reported in the separate IPCC reports. SR1.5 categories "1.5C low overshoot" and "Below 1.5C" have been mapped as C1, "1.5C high overshoot" as C2, "Lower 2C" as C3, and "Higher 2C" as C4.**