# Peer review of "The IPCC Sixth Assessment Report WGIII climate assessment of"

_EGUsphere, 2022_

## Author Comment (AC2)

**The IPCC Sixth Assessment Report WGIII climate assessment of mitigation pathways: from emissions to global temperatures**

Jarmo S. Kikstra, Zebedee R. J. Nicholls, Christopher J. Smith, Jared Lewis, Robin D. Lamboll, Edward Byers, Marit Sandstad, Malte Meinshausen, Matthew J. Gidden, Joeri Rogelj, Elmar Kriegler, Glen P. Peters, Jan S. Fuglestvedt, Ragnhild B. Skeie, Bjørn H. Samset, Laura Wienpahl, Detlef P. van Vuuren, Kaj-Ivar van der Wijst, Alaa Al Khourdajie, Piers M. Forster, Andy Reisinger, Roberto Schaeffer, and Keywan Riahi

*In review at Geoscientific Model Development*

Version history

| | |
|---|---|
| 28 June 2022 | Original submission |
| 24 August 2022 | Open review period closed – final response period starts. |
| 21 September 2022 | Submission author response to review comments |

Response to reviewers' comments

Reviewers' comments are shown in black text.

We use red text to denote our responses to the comments.

We use indented blue text for relevant sections of the revised manuscript and SI.

**Reviewer report #1**

This paper describes a key set of processes used by the Intergovernmental Panel on Climate Change to estimate climate outcomes from emissions scenarios. The text is well- written. As such it is worthy of publication.

We thank the reviewer for their positive comments.

I do however have some comments and proofing points for the authors to consider, as follows:

[R1.C1]

There is no discussion as far as I can tell of the impact of harmonisation and infilling at the regional scale. Basing these steps on global values (of CO2) implicitly assumes that the ratio of CO2 to other species is homogeneous across space, which is unlikely to be true. What are the implications of this workflow for use of the regional emissions data in the AR6 database?

Most of the climate mitigation pathways reported in the scenario database used by WGIII AR6 also have regional emissions data. However, the climate emulators applied by WGIII for the climate simulations are relatively simple - with the primary aim here being the assessment of global warming outcomes. One simplification that follows is that they generally run based on global emissions, rather than on regional emissions. Considering the primary aim of this application, and noting that regional emissions processing would introduce further uncertainties not yet explored for the infilling step, working on the level of regional emissions would be beyond the scope of this work. The entire workflow is therefore done on a global level.

Similar to previous work, such as in the IPCC Special Report on 1.5C, if one would want to link regional emissions to global climate outcomes, care must be taken, because the regional detail has not been used directly. We recognise that, for instance, different regional aerosol emissions lead to different regional climates. Some text will be added to the manuscript in the introduction and the discussion to recognise this and point out that the comparison of global aggregate and regional emissions processing would be worthwhile looking into for future studies.

> "While most of these scenarios contain regional emissions pathways, WGIII AR6 only assessed global climate variables based on global emissions estimates, which is the common level that the used climate emulators operate on. This means that evaluating the regional effects of for instance regional aerosol emissions is beyond the scope of this assessment, having as a primary aim the assessment of global mean surface temperature change." (introduction)

Additionally, it may be worth pointing out a bit more clearly that while the AR6 database does feature regional emissions for most of the 1202 global scenarios assessed, it only reports global climate variables. In section 5.1.3 of the article, a discussion on extending the climate assessment workflow to regional detail already existed, but the text did not yet explicitly mention regional emissions. We can add some text to be more explicit:

> "For instance, the current climate assessment workflow from emissions to global temperature change could be extended to enable the inclusion of regional emissions detail, and their effect on regional climate. This could for instance come in the form of emulator to emulate regionally downscaled climate change projections, using tools such as MESMER (Beusch et al., 2020, 2022), or other modelling approaches that utilise regional emissions

data available in the AR6DB to enable differentiation between for instance regional aerosol emissions pathways." (discussion)

[R1.C2]

This is a question so naive that I am slightly embarrassed to ask... Are the emissions data in the AR6 database the unadjusted submitted data, or are they adjusted data (harmonised, infilled, or both)? If unadjusted, then users should exercise caution in estimating emissions-climate properties (such as TCRE) from the database. If adjusted, users should know they do not necessarily match the submitted data and may exhibit biases from processing. Either way, this should be clear in the database metadata (e.g. https://data.ece.iiasa.ac.at/ar6/#/about) and in the paper. Perhaps it is and I am unobservant, but I did spend a while looking!

Thanks for this comment. It is always useful to know where things are unclear - especially for careful readers like reviewers! In the database (AR6DB), there are a few different types of emissions variables. "Emissions|*" are the unadjusted submitted data. Every variable that has "AR6 climate diagnostics" in the name is adjusted by the workflow described in this paper. "AR6 climate diagnostics|Harmonized|Emissions|*" are the emissions trajectories after the harmonisation step. "AR6 climate diagnostics|Infilled|Emissions|*" are the emissions trajectories after both the harmonisation and infilling processes. In the database (AR6DB) you will also find a few variables called "AR6 climate diagnostics|Native-with-Infilled|Emissions|*", which are based on unadjusted submitted emissions trajectories where available, and otherwise supplemented by infilled ("AR6 climate diagnostics|Infilled|Emissions|*") variables.

We will add this information to the supplementary material of this manuscript, and make sure the AR6DB description will be updated with this information. Together with the to-be-submitted revision, the database will see an update that should have all this necessary information and data made available.

[R1.C3]

What is the justification for the scenario selection criteria? And what is the sensitivity of the number of included scenarios to variations in these criteria? The selection criteria are described briefly in lines 300-310 and listed in Supplementary Figure 1, but no reasoning for these criteria or the threshold values is given, either in this manuscript or in IPCC WG3 AR6 Annex 3.

Selection of the criteria was devised iteratively over several meetings amongst WGIII authors, incorporating information about different uncertainties in historical values and expert judgement. In this first version of the manuscript, we chose to not describe the vetting process in too much detail, because this choice can be seen as somewhat 'external' to the climate assessment workflow from emissions to temperatures. Seeing the questions from reviewers however (here, and comment R2.C4), we will extend the discussion around it by adding a description of the process and justification, in the supplementary information.

[R1.C4]

Table 1 appears to show harmonisation methods only, not infilling methods as suggested by the caption. It would be good to see the infilling procedures included in the table - perhaps as an extra column?

Table 1 includes two columns that pertain to infilling, titled "Infilling Method" which specifies the algorithm applied, and the "Infiller database" used to derive the infilled pathways from, for each gas in question. We will expand the caption of Table 1 to increase clarity, with some text along the lines of:

> "The database used for harmonisation was in all cases the database also used for RCMIP [Nicholls et al. 2021]. The reason for varying the infilling method and database are explained in section 3.3 of the text of this manuscript, and is purely dependent on the availability of the number of modelled pathways and their independence in each database. QRW is used when a sufficient number of independent pathways is available in the AR6 database, otherwise RMS-closest is chosen. CMIP6-SSPs is chosen as the database if the gas in question is not represented in the AR6 database."

[R1.C5]

Lines 561-564: while I think the authors are correct that sensitivity to absolute warming can act as a proxy for sensitivity to uncertainty in a bundle of other factors, the relationships are not intuitive. The reader would benefit from a brief explanation of how to infer from Supplementary Figure 1 the implications for uncertainty in forcing or of harmonisation & infilling emissions.

We thank the reviewer for this point. To address it, we will add some extra text the manuscript along the lines of:

> "This simple sensitivity analysis on the level of global temperatures gives a sense for how much scenario categorisation is related to uncertainty in climate projections of emissions pathways. This can be connected to the change in categorisation that may come with a potential change in harmonisation and infilling methods, but it is not immediately obvious what effect a change in harmonisation or infilling would have on categorisation. In section 4.7 of this article, we discuss the temperature change that can be attributed to changes in climate assessment methods between SR1.5 and AR6, therewith providing an initial analysis by showing the magnitude of the changes between the two applications. However, a full analysis of the uncertainties in the climate assessment workflow is beyond the scope of this paper and remains a topic for further research."

[R1.C6]

In Figure 7, can the authors provide an additional panel showing the impact of infilling and harmonisation on other forcings beyond Kyoto Gases? These are also important, and perhaps where some of the largest (proportional) changes arise from infilling and harmonisation.

We thank the reviewer for this comment, and concur that it would be helpful to add more information on non-Kyoto Gases emissions changes resulting from harmonisation and infilling. In a revised version, we will add information on the median change over time of a few key non-GHG emission species, by replacing the somewhat redundant "Rest of Kyoto Gases" in panel D (the same information can be easily derived from panels A-C) and replacing it by the effect of infilling and harmonisation on organic carbon (OC), black carbon (BC) and sulfur emissions. We will add some text that describes the panel as well.

> "The emissions processing also affects climate forcers beyond the Kyoto Gases, which are not readily expressed in GWP100 CO2-equivalent values. Most evaluated scenarios model non-Kyoto climate forcers such as black carbon (BC), organic carbon (OC) and sulfur, but the relative difference in reported past emissions can be quite large (Figure 7D)."

And a few proofing suggestions:

- Line 75: italicise "climate"?

  Thanks. We will include this suggestion.

- Line 112: missing "the" between "of" and "limited".

  Thanks. We will modify as follows:

  > "Because of the limited ability in the 1990s to perform"

- Lines 466-467: this sentence needs completing.

  Thanks. We will update the sentence to:
  > "Inspired by Geden and Löschel, (2017) and recent scenario studies investigating temperature overshoot (Drouet et al., 2021; Riahi et al., 2021; Johansson, 2021; Tachiiri et al., 2019), we add an analysis of the overshoot severity of all assessed pathways of the AR6WGIII report as the cumulative years above a certain global warming level, multiplied by the projected average annual climatic °C overshoot in each year."

- Table 1 caption: A ** footnote is explained, but I don't see that symbol used in the table.

  We have removed the footnote, which was accidentally left in the caption.

- Figure 5, panels B-D: Could the authors explain in the caption how the scenarios have been ordered? Would it be informative to order them by magnitude of ODY_1.5, for instance?

  The ordering was done alphabetically. We agree that this would be a missed opportunity where more information can be included. Some options for ordering would be the onset year of overshoot, the duration of the overshoot, the peak level of overshoot, and indeed the overshoot degree years. We like the reviewer's suggestion and will update the figure to be ordered by peak $ODY_{1.5}$, which fits well with the following panels in this figure.

**Reviewer report #2**

This paper details the climate assessment processing steps developed between IPCC Working Groups I and III to assess temperature outcomes of emissions pathways and introduces an open source climate assessment Python package that can facilitate this processing. The processing steps include (1) vetting of emissions scenarios submitted to the AR6 Scenario Explorer, (2) harmonization of vetted emissions scenarios with historical emissions, (3) infilling of missing species in the emissions scenario, and (4) assessment of the emissions-climate response with three climate emulators. The paper then evaluates the impact of this processing on global-mean temperature projections and effective radiative forcing statistics before analyzing overshoot degree years and 'Paris-compatible' emissions scenarios.

[R2.C1]

The paper is well-written and provides a valuable reference for elucidating the IPCC scenario-emissions process. I would recommend publication with a few qualifications below to better emphasize the novelty of this contribution (for instance fore-fronting the statement in lines 245-248).

We thank the reviewer for their positive and constructive comments. We moved the suggested sentence to the first paragraph.

[R2.C2]

This paper read to me as part review/part analysis/part overview of a new community tool, which is a lot to take on but is very useful to the climate community at large. My recommendation would be to highlight the community tool more clearly throughout the methods section to encourage its uptake. This could include providing a schematic of the workflow either in the main text or supplement or clarifying throughout the methods section how and where the *climate assessment* workflow package was used. I'll note that while the workflow illustrated in Figure 1 is excellent, it is more supportive of the review portion of the text as opposed to elucidating the packages, emulators, and analysis included in the community tool.

We thank the reviewer for their comments and would concur that this paper serves a few different purposes. In the revision, we are very happy to follow the suggestion of adding a visualisation with a short description in the text that highlights the work here more as a community tool, which will hopefully also help placing it within the existing efforts and provide suggestions for future work. This visual will then be connected to several points already made in the discussion section, pointing out current functions and possible extensions of this workflow.

[R2.C3]

Why aren't CICERO-SCM results reported in the abstract? Or make it clearer that it is used only for sensitivity analysis.

CICERO-SCM was only used for sensitivity analysis due to some identified limitations like the lack of an interactive carbon cycle, and projecting lower warming than the best assessment along SSPs. In the revised manuscript this will be made more clear, including in the abstract.

[R2.C4]

Suggestion: In addition to Table 2 in the Supplementary file, a bar graph displaying the vetting 'success' of scenarios from each model would be useful. It would emphasize the number of scenarios that are included by a disproportionate number of IAMs (as discussed in section 5.1.1).

Thanks for this suggestion. We will include information from AR6WGIII Ch.3 and Annex III that shows the number of the submitted scenarios versus the number of scenarios that passed the vetting requirements, by model framework. This will then come alongside a short discussion on the justification for the vetting specifically with regard to the application here, i.e. the derivation of consistent global average surface temperatures based on the emissions trajectories of each scenario (see our answer to reviewer comment R1.C3).

[R2.C5]

It is exciting to consider the use of emulators for variables beyond global estimates of GSAT, but other variables have not yet been comprehensively evaluated, such as precipitation. Should there be some discussion of the uncertainty and potential of emulators to provide societally relevant metrics beyond GSAT? Similarly, more discussion of the regional emissions data vetting and application would be useful as a means of underpinning the recommendation in the concluding sentence.

In the revised version we will include a list of climate variables that are the output from the *climate-assessment* package. While this includes a few variables not explored in detail in this work or AR6WGIII, it does not include for instance precipitation. The climate emulators applied in this effort are not able to simulate regional or local climate outcomes and do not include precipitation. We therefore would argue that such a discussion is beyond the scope of this manuscript, and would leave it at the discussion in 5.1.3, where we clarify that precipitation and other societally relevant indicators (and any quantification of its uncertainty) is part of future research.

And a few proofing suggestions:

- Line 300: delete in: '…climate mitigation options [in] were extensively…'
  Okay. Thanks for spotting this.
- Line 490: Chen et al. 2021 not included in the references
  Okay. Thanks for spotting this.
- Style: Remove the indent from lines 219, 223, 226, 228
  Okay. Thanks for spotting this.
- Line 252: "A growing body of research has been developed to describe[ing] analyses that compare…" (Or something along these lines)
  Okay. Thanks for spotting this.
- Line 296: italicize 'climate assessment' as is done on line 814
  Okay. Thanks for spotting this.
- Line 300: delete 'in': "Global scenarios used to assess climate mitigation options [in] were extensively…"
  Okay. Thanks for spotting this.
- Line 444: delete 'above': "Beyond these, AR6 WGIII includes categories [above] relevant for higher emissions scenarios that…"
  Okay. Thanks for spotting this.

- Line 765: delete 'a': "While overshoot indicators like $ODY_5$ may immediately be [a] useful as an…

  Okay. Thanks for spotting this.
- Supplement: Include the number of scenarios considered in Table 2.

  Thanks for the suggestion. We will take this up together with comment [R2.C4], which asked us to show the number of scenarios submitted versus the number that passed the vetting.

---

## Author Response (AR2)

**Topical Editor, comments to the author (11 Nov 2022)**:
Great job responding to most of the reviewers' comments. However, your responses to several reviewer comments (R1.C2, R1.C3, R2.C4) indicate that changes would be made to the Supplementary Material, but I don't see them. Please make those changes and upload the updated Supplement, including a tracked-changes version.

There are some results that have changed. Why? E.g. (line numbers in tracked-changes version), L43, L900-10 .

Non-public comments to the Author:
Minor things (line numbers in tracked-changes version):

L333: Add "a" before "growing"

L807-21, also R1.C5: Please include references to sub-plots (e.g., 1a), which would help reader understanding of these figures.

L1634: Should "were" be "were not"?

**Author responses alongside new upload (15 Nov 2022):**

However, your responses to several reviewer comments (R1.C2, R1.C3, R2.C4) indicate that changes would be made to the Supplementary Material, but I don't see them. Please make those changes and upload the updated Supplement, including a tracked-changes version.

Thank you for noticing that these additions were still missing! Now we have added more information in the Supplementary Material, which we belief should be enough to satisfy these three reviewer comments.

We have uploaded:

- This response (v2_0_2-response_to_editor)
- An updated supplement (v2_0_2)
- Tracked changes supplement, from v1 to v2_0_2
- An updated manuscript (v2_0_2)
- Tracked changes supplement, from v2 to v2_0_2

More specifically, for R1.C2 we now added Supplementary Tables 3-5, which are detailed lists of the emissions and climate variables related to the climate assessment workflow, with references to it in the text. To follow up on R1.C3, we have now added a paragraph in the Supplementary Material on vetting and refer to that in the main manuscript. For R2.C4, we had originally hoped that the addition of Supplementary Figure 2 and the changes to Supplementary Table 2 (the inclusion of totals by category and model framework) would be enough – but we realise that in line with the answer to R1.C2 we should have added some text. Now, indeed in line with the missing information that was promised in response to R1.C3, we added text to further explain Supplementary Figure 2 and related to the vetting explanation.

There are some results that have changed. Why? E.g. (line numbers in tracked-changes version), L43, L900-10.

> That is correct. The changes fall in two categories.
>
> - *Update to v1.1 scenario database = consistency with the IPCC report.*
>   The AR6 public database update v1.1 now includes a file with more detail on the climate variables, including the FaIR and CICERO data. It was noticed that in our initial submission, a few bugs were still left in the scenario data, which resulted in not having data for all scenarios in all calculations. These have now been resolved to (a) be fully in line with the correct number of scenarios in the IPCC report and (b) have the same handling of how to calculate the Kyoto Gases basket.
>   In short, the main bug was that in the previous version of the data used for the preprint, a string matching error occurred. due to the scenario string of a handful of scenarios from the FaIR data had one non-capitalized letter, which was capitalized in the MAGICC and CICERO data. This caused the matching to fail, leaving those scenarios out of the comparison. We found this mistake and fixed it, with the number of scenarios (1202) with temperature assessment now everywhere being in line with the IPCC report.
>   More specifically:
>
>   - L43; this was an accidental bug in the first version, where some scenarios were accidentally dropped in the handling of the v1.0 of the detailed database. This did not affect the main database, but was caused by some differences in capitalized/non-capitalized scenario name strings in the climate data of FaIR and CICERO which was different from that of MAGICC, causing that these were not successfully combined. Now, our results are back in line with the IPCC report again, after this inconsistency in the extra climate data was resolved.
>     This can be seen perhaps most clearly in the updated numbers of scenarios in Figure 3B.
>
>   - L900-10: The scenario numbers here were caused by a similar issue, where the extra climate data used in v1 was not exactly in line with what was used in the IPCC report which has now been resolved.
>
> - *Updated (clearer) language.*
>   Any other changes only result from an attempt to provide slightly more clear communication about the insights. The one change here to highlight would be line 884 (in tracked changes), on the high end of F-gases emissions. We switch the language from 7Gt/yr to 5Gt/yr. While the previous version with 7Gt/yr was true, it relied on only one scenario in the database. To avoid giving the possible impression that 7Gt/yr is a common factor in the AR6DB, we opted to go with more robust language, changing it to 5Gt/yr in combination with "a set of high emissions scenarios".
>
> It should be noted that these changes were marginal, and no insights have changed fundamentally.

L333: Add "a" before "growing"

Thanks, done.

L807-21, also R1.C5: Please include references to sub-plots (e.g., 1a), which would help reader understanding of these figures.

We have now added references to sub-plots of Supplementary Figure 1, in the main text.

L1634: Should "were" be "were not"?

Using "were" is correct here. We have added "… such that infilling was not necessary." to clarify.